# Distilled Gradient Aggregation:
# Purify Features for Input Attribution
# in the Deep Neural Network

**Giyoung Jeon**[1,3*] **Haedong Jeong**[1,2*] **Jaesik Choi**[2,3]

[1]UNIST [2]KAIST [3]INEEJI

giyoung@unist.ac.kr haedong.jeong@kaist.ac.kr jaesik.choi@kaist.ac.kr

## Abstract

Measuring the attribution of input features toward the model output is one of the popular post-hoc explanations on the Deep Neural Networks (DNNs). Among various approaches to compute the attribution, the gradient-based methods are widely used to generate attributions, because of its ease of implementation and the model-agnostic characteristic. However, existing gradient integration methods such as Integrated Gradients (IG) suffer from (1) the noisy attributions which cause the unreliability of the explanation, and (2) the selection for the integration path which determines the quality of explanations. FullGrad (FG) is an another approach to construct the reliable attributions by focusing the locality of piece-wise linear network with the bias gradient. Although FG has shown reasonable performance for the given input, as the shortage of the global property, FG is vulnerable to the small perturbation, while IG which includes the exploration over the input space is robust. In this work, we design a new input attribution method which adopt the strengths of both local and global attributions. In particular, we propose a novel approach to distill input features using weak and extremely positive contributor masks. We aggregate the intermediate local attributions obtained from the distillation sequence to provide reliable attribution. We perform the quantitative evaluation compared to various attribution methods and show that our method outperforms others. We also provide the qualitative result that our method obtains object-aligned and sharp attribution heatmap.

## 1   Introduction

Deep Neural Networks (DNNs) are increasingly applied to many fields in human-life such as self-driving, medical predictions and time-series forecasts. Along with these improvements, the recent models get bigger and more complicated that humans cannot investigate and understand the internal decision mechanism of them. Identifying and analyzing the reasons for the model predictions are important because the malfunction or the groundless decision of the model can cause the critical problems. As an effort to provide the evidences on the decisions, the input attribution has been well-studied, especially in visual tasks. Input attribution method aims to measure how much each input feature contributes to the model prediction. Because the output of this method takes the form of heatmap to provide relative importance among input features, it helps to locate the appearance of human-level semantics in the input. Previous work also applies the attribution to debug the model by wiping out Clever Hans of DNNs [Lapuschkin *et al.*, 2019]. However, obtaining the trustful input attribution is still challenging because (1) the highly nonlinear structure of modern DNNs makes it difficult to correctly track the relationship between the input and the output, and (2) quantifying the reliability of attribution methods is non-trivial because the ground-truth is not available.

---

*Equal Contribution

36th Conference on Neural Information Processing Systems (NeurIPS 2022).

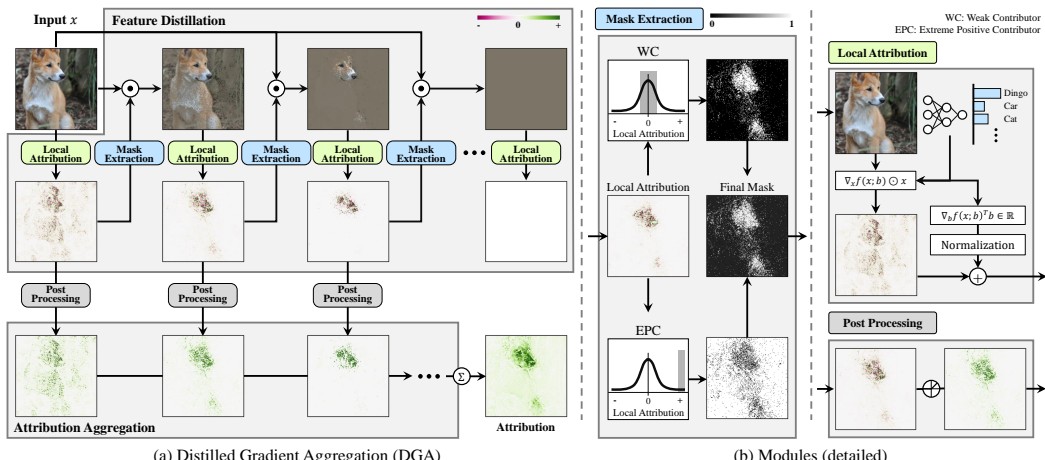

| (a) Distilled Gradient Aggregation (DGA) | (b) Modules (detailed) |

Figure 1: An illustration of Distilled Gradient Aggregation (DGA) and details of modules included in the computation of DGA. DGA generates the discrete sequence of anchor points by distilling the input features. We identify that aggregating the local attributions along the distillation sequence provides the class (*Dingo*) oriented attribution heatmap. The details are described in the Section 3.2

Gradient-based input attribution is one of the main techniques to derive the relationship between the model decision and the input features. The partial derivative of the output with respect to the input provides the measure of sensitivity, which is easily computed in DNNs. Integrated Gradients (IG) [Sundararajan *et al.*, 2017] is a commonly used gradient-based method, which provides the axiomatic properties to support the reliability of attributions. However, IG inheres the problem of noisy attribution, which originates from the gradient integrating path, and several variants of IG have been proposed to alleviate this issue [Smilkov *et al.*, 2017; Kapishnikov *et al.*, 2019, 2021; Pan *et al.*, 2021]. FullGrad (FG) [Srinivas and Fleuret, 2019] also raises the counter-intuitive behaviors of IG. FG avoids this problem by considering only the local gradients instead of the path integration and proposes to use the bias gradient. But FG is vulnerable to the small perturbation in the inputs due to its locality.

In this work, we provide the analysis on the weakness of (1) FG method which is unavoidable if it considers only single anchor point, and (2) IG method which the continuous path based gradient integration may fail to quantify the intuitive attribution. To complement the shortcomings of the two methods, we propose to aggregate the attribution from the multiple anchors. For the selection of anchor points, we devise an algorithm to sequentially distill the irrelevant features to generate the reliable attribution. The main contributions of our work are,

- Propose a novel feature distillation algorithm based on the intermediate local attribution to generate the sequence of meaningful anchor points.
- Devise Distilled Gradient Aggregation (DGA), an attribution method by aggregating the intermediate local attributions from the distilled input sequence for the reliable attributions.
- Qualitative and quantitative evaluations to validate the proposed method outperforms existing gradient-based attribution methods.

## 2  Related Work

Input attribution is one of the post-hoc explanation methods, which aims to identify the influence of each input feature to the model output on the trained model. There exists a considerable variety of techniques to derive the input attribution. With the property that the feature map obtained by the convolutional layers includes the spatial information, Class Activation Mapping (CAM) methods compute the attribution by the weighted sum of the feature maps [Zhou *et al.*, 2016; Selvaraju *et al.*, 2017]. Layer-wise Relevance Propagation (LRP) method propagates the model output backward to the input [Bach *et al.*, 2015; Nam *et al.*, 2020]. LRP extends the Taylor decomposition to the DNNs and distributes the relevance in layer-wise sense.

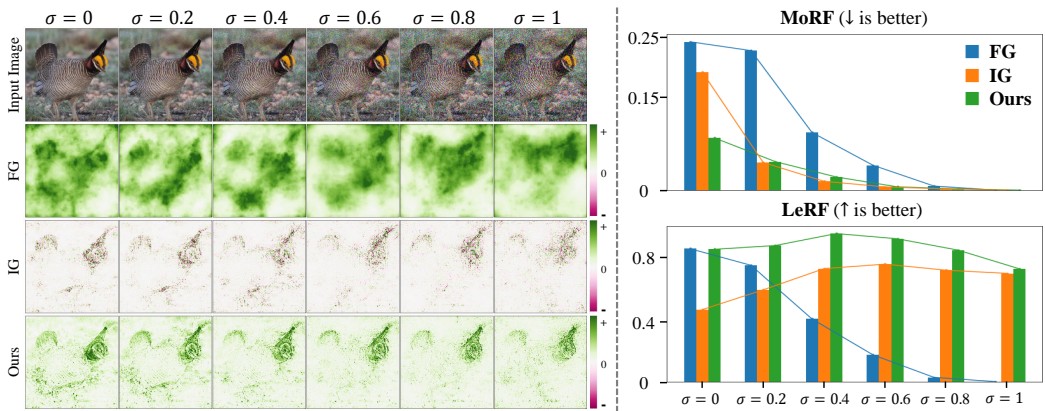

Figure 2: Attribution heatmaps for the Gaussian noise perturbed images obtained by FullGrad (FG), Integrated Gradient (IG) and Ours. Heatmaps from FG shows inconsistent results while other two methods are consistent, which utilize the global perspective for the attribution. Corresponding pixel perturbation scores also show that FG loses reliability against the simple noise perturbation. The procedure of pixel perturbation is described in Section 4.1

There are another approach by measuring the behavior of the model by perturbing the input features. Optimizing the input by gradient ascent gives an example which would maximally activates the target neuron [Erhan *et al.*, 2010; Nguyen *et al.*, 2016; Olah *et al.*, 2018]. Instead of maximizing the target neuron activation, Extremal Perturbation optimizes the mask which removes or reveals the part of the input to localize the attributed part of input [Fong *et al.*, 2019]. It is extended by using Integrated Gradients [Sundararajan *et al.*, 2017] for the optimization to make the optimization more stable [Qi *et al.*, 2020]. By collecting the pair of partially removed inputs and corresponding model outputs, training a linear model to resembles the mapping would give the feature importance in terms of linear weights [Ribeiro *et al.*, 2016]. Rather than training a new model, Randomized Input Sampling for Explanation (RISE) computes the attribution by aggregating multiple randomly masked inputs, weighted by the model outputs [Petsiuk *et al.*, 2018].

Based on Aumann-Shapley value [Aumann and Shapley, 2015], which is one solution of the fair distribution in the cooperative game theory, Integrated Gradients (IG) has been proposed [Sundararajan *et al.*, 2017]. IG is equipped with axiomatic properties which are desirable for the attribution methods. IG is computed by integrating gradients over the straight path from the predefined baseline to the input. As the attribution is corrupted by the noisy information along the path, alternatives for the different paths have been proposed [Smilkov *et al.*, 2017; Kapishnikov *et al.*, 2019, 2021]. FullGrad (FG) [Srinivas and Fleuret, 2019], which utilizes the bias-gradient, is proposed to suppress the counter-intuitive behavior of IG which is caused by the weak-dependency between the local linear regions.

## 3 Distilled Gradient Aggregation Method

In this section, we propose our gradient aggregation method, Distilled Gradient Aggregation (DGA). We first provide an example and analysis about the inconsistency observed by FullGrad, which uses a single anchor point to compute attributions. We also provide the counter-intuitive behavior of IG caused by the continuous gradient integration path. To complement both shortcomings, we propose an aggregation method, which ensembles the local attribution from the sequence of inputs. To reduce computational cost and reinforce the features that are in charge of the model decision, we suggest a sequential feature distillation algorithm, which distills irrelevant features from the input.

### 3.1 Analyzing FG and IG on Simple Models

Assume we have the input vector $\mathbf{x} \in \mathbb{R}^2$ and a simple neural network $f$ equipped with partial linear activation (e.g., ReLU). This network $f$ can be regarded as the combination of piece-wise linear functions [Montufar *et al.*, 2014]. Each piece-wise linear function is only defined and feasible in

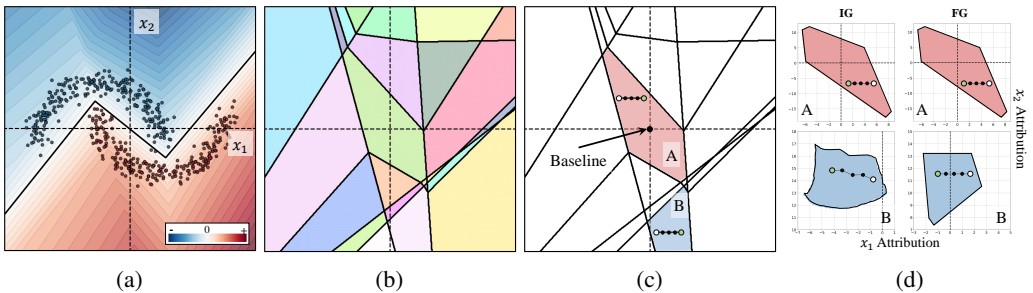

Figure 3: (a) A contour of logit values for trained $f$. (b) The linear regions which the trained network comprises. Each colored linear region corresponds to each piece-wise linear function. (c) Selected two linear regions (A and B) and the zero baseline. The dotted lines indicate the perturbations for $x_1$ axis in the same linear region. (d) Attribution of IG and FG for each linear region. We identify that for linear region A (include the baseline), the global attribution (IG) is same as the local ones (FG). However, for linear region B, the global and local attribution has different attributions for input samples.

corresponding linear region $\mathcal{R}^{(k)}$, where $\cup_k \mathcal{R}^{(k)} = \mathbb{R}^2$ and $\mathcal{R}^{(k_1)} \cap \mathcal{R}^{(k_2)} = \varnothing$ for any $k_1$ and $k_2$. Such piece-wise linear function is formulated as follow,

$$f(\mathbf{x}) = \begin{cases} \mathbf{w}^{(1)T}\mathbf{x} + \mathbf{b}^{(1)} & \mathbf{x} \in \mathcal{R}^{(1)} \\ \cdots \\ \mathbf{w}^{(K)T}\mathbf{x} + \mathbf{b}^{(K)} & \mathbf{x} \in \mathcal{R}^{(K)} \end{cases} \tag{1}$$

where $\mathbf{w}^{(k)} \in \mathbb{R}^2$ and $\mathbf{b}^{(k)} \in \mathbb{R}$ denote weight and bias of $k$-th linear region respectively. Figure 3 depicts an illustrative example of the function $f^2$. For the network $f$, FullGrad (FG)[Srinivas and Fleuret, 2019] and Integrated Gradient (IG)[Sundararajan *et al.*, 2017] are given as follow,

$$FG(\mathbf{x}) = \Psi(\nabla_{\mathbf{x}} f(\mathbf{x}) \odot \mathbf{x}) + \sum_{l \in L} \sum_{c \in \mathbf{c}_l} \Psi(\nabla_{b_c} f(\mathbf{x}) b_c) \tag{2}$$

$$IG(\mathbf{x}) = \int_{\alpha=0}^{1} \nabla_{\gamma(\alpha)} f(\gamma(\alpha)) \odot \nabla_{\alpha} \gamma(\alpha) d\alpha \tag{3}$$

**Vulnerability of FullGrad** FullGrad suggests that the attribution should be same inside the same linear region $\mathcal{R}^{(k)}$, and this reduces the dependency between the attribution and the input $\mathbf{x}$. This property is introduced as *weak dependency*. However, such weak dependency derives the attribution to be vulnerable to the model perturbation. For example, let we have two inputs, $\mathbf{x}$ and $\mathbf{x}' = \mathbf{x} + \epsilon$, where $\epsilon$ be any small enough random perturbation. If we find any $\mathbf{x}$ and $\mathbf{x}'$, such that the model output is same, $f(\mathbf{x}) = f(\mathbf{x}')$, but the region is different, then the attribution on each input should be different. This can be visualized by the simple experiment by generating noise perturbed image $\mathbf{x} + \epsilon$ and measuring the attribution, where $\epsilon \sim N(0, \sigma I)$. Figure 2 shows an example where the FullGrad generates inconsistent attribution along with the simple Gaussian noise is added.

**Counter-intuitive behavior of IG** To visualize the counter-intuitive behavior of IG, we select two linear regions ($A$, $B$) in Figure 3c and calculate the attribution in each region. In particular, we select a sequence of data from $a$ (white dot) to $b$ (green dot), which is only shifted in $x_1$ dimension. Figure 3d illustrates corresponding attribution for two selected linear regions. We observe that only attribution of $x_1$ changes for the sequence of region $A$ in both IG and FG methods. However, for the sequence of region $B$, the IG attribution of both $x_1$ and $x_2$ changes at the same time, while FG attribution shows attribution change only in $x_1$. We conjecture this counter-intuitive behavior of IG is caused by the baseline selection. With the zero-baseline ($\bar{\mathbf{x}} = 0$), the integration paths of samples in region $A$ traverse only a single region to compute IG. On the other hand, paths of samples in region $B$ traverse through multiple regions. When traversing multiple regions, the counter-intuitive behavior can be induced due to passing the undesirable linear regions. From this observation, we can identify that the selection of baseline determines (1) which linear regions are traversed by the

---

[2] The implementation details are described in the Appendix J.

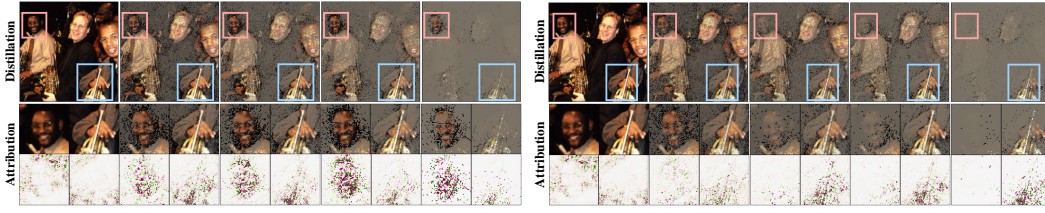

(a) The sequence $\tilde{\mathcal{X}}$ with WC mask.   (b) The sequence $\tilde{\mathcal{X}}$ with WC and EPC mask.

Figure 4: Distillation sequence $\tilde{\mathcal{X}}$ with WC mask and EPC mask for the target class *French horn* in the pre-trained VGG-16. The bottom row of (a) and (b) indicates the local attribution $\phi^{UFG}(\tilde{\mathbf{x}}(n))$ for each colored box of first row.

path, and (2) how much portion of the path is included in each selected linear region. Although the proper selection of baseline can be an one option to adjust (1) and (2) for the reliable attributions, it is still non-trivial to control the sequence of the meaningful linear regions and each weight by only changing the baseline.

Based on these insights, we raise the weakness of the local and global attribution methods: (1) vulnerability and (2) counter-intuitive behavior. To alleviate these issues, we desire an attribution method to combine the strengths of each type of attribution methods. To resolve such desire step by step, we start from the local attribution which is free from the baseline selection. Then, the remained problem is how to select the meaningful linear regions to generate the reliable attributions.

### 3.2 Sequential Feature Distillation

For the strategy to select linear regions, RISE [Petsiuk *et al.*, 2018] suggests the random perturbation-based approach. RISE explores the multiple linear regions with randomly ablated masks to measure the importance of each ablated features. However, the randomized ablation includes the stochastic process which requires expensive computational cost to achieve reliable attributions. We are also inspired from the adaptive selection for the perturbed inputs in Guided IG (GIG) [Kapishnikov *et al.*, 2021], which can improve the final attribution. Thus, we hypothesize that the adaptive exploration of linear regions based on the intermediate local attribution can reduce the cost of randomized exploration and follow the spirit of the adaptive exploration. Finally, we propose the sequential feature distillation algorithm to obtain a sequence of ablated inputs, the sequence of inputs $\tilde{\mathcal{X}} = [\tilde{\mathbf{x}}(0), \tilde{\mathbf{x}}(1), \tilde{\mathbf{x}}(2), \cdots]$, where the irrelevant features are distilled. Motivated by previous work that masking out the irrelevant features using IG [Fong *et al.*, 2019; Qi *et al.*, 2020], we propose to distill the impurities by using the intermediate local attribution obtained along the sequence.

For reliable local attribution, FG proposes the bias gradient with post-processing $\Psi(\cdot)$ which includes the normalization and upsampling. As $\Psi(\cdot)$ suggested by FG is usually over-estimated by the bias gradient in deeper layers [Grabska-Barwinska *et al.*, 2021], we redefine $\Psi(\cdot)$ as uniformly distributing function for the bias gradient to alleviate the over-estimation problem.

$$\Psi_u\left(\mathbf{v}\right) = \frac{\mathbf{v}^T \mathbb{1}^{\dim(\mathbf{v})}}{\dim(\mathbf{x})} \mathbb{1}^{\dim(\mathbf{x})} \tag{4}$$

where $\mathbb{1}^d$ denotes a $d$-dimensional all-ones vector. We call FG with redefined post-processing $\Psi(\cdot)$ as Uniform FullGrad (UFG), $\phi^{UFG}(\cdot)$. Then we use UFG as the intermediate local attribution method throughout the remained paper. We note that different local attribution, such as Grad*Input [Shrikumar *et al.*, 2016], can be used in our method. We provide the comparison of selecting different local attribution in Appendix G.

To distill off the uninformative features, we build a mask to zero out the features with low magnitude of local attribution. In the sequence $\tilde{\mathcal{X}}$, the relation between $n$ and $n+1$-th ablated input is formalized as,

$$\tilde{\mathbf{x}}(n + 1) = \mathcal{M}(\tilde{\mathbf{x}}(n)) \odot \tilde{\mathbf{x}}(0) \tag{5}$$

where $\tilde{\mathbf{x}}(0) = \mathbf{x}$ and $\mathcal{M}(\cdot)$ is a mask extractor. We define this mask extractor as the Weak Contributor (WC) mask, $\mathcal{M}^{WC}$. The level of the WC mask increases along the distillation sequence $\mathcal{X}$ with the

---
**Algorithm 1** Distilled Gradient Aggregation
---
**Input**: Model $f$, Input $\mathbf{x}$
**Parameter**: # of steps $N$, EPC threshold $q$, Negative scale $\beta$
**Output**: Attribution $\phi^{DGA}(\mathbf{x})$
 1: Let $\tilde{\mathbf{x}}(0) = \mathbf{x}$, $\Phi = \emptyset$
 2: **for** $n$ in $\{0 \ldots N\}$ **do**
 3:     $\Phi = \Phi \cup \{\phi^{UFG}(\tilde{\mathbf{x}}(n))\}$
 4:     $\mathcal{M} = \frac{n}{N}\mathcal{M}^{WC}(\tilde{\mathbf{x}}(n), n; N) + (1-\frac{n}{N})\mathcal{M}^{EPC}(\tilde{\mathbf{x}}(n); q)$
 5:     $\tilde{\mathbf{x}}(n+1) = \tilde{\mathbf{x}}(0) \odot \mathcal{M}$
 6: **end for**
 7: $\phi^{DGA}(\mathbf{x}) = \sum_{\phi \in \Phi} \text{ReLU}(\phi)$
 8: **return** $\phi^{DGA}(\mathbf{x})$
---

pre-defined number of steps $N$ and finally the entire pixels become zero (i.e., $\tilde{\mathbf{x}}(N) = 0$). We define $\mathcal{M}_j^{WC}(\cdot)$ for each feature $j$ as,

$$\mathbb{S}_j^{WC}(\mathbf{x}) = \left\{ k \middle| \ |\phi_k^{UFG}(\mathbf{x})| \leq |\phi_j^{UFG}(\mathbf{x})| \right\} \tag{6}$$

$$\mathcal{M}_j^{WC}(\mathbf{x}, n; N) = \begin{cases} 0 & \text{if } \frac{|\mathbb{S}_j^{WC}(\mathbf{x})|}{dim(x)} \leq \frac{n}{N} \\ 1 & \text{otherwise} \end{cases} \tag{7}$$

where $\mathbb{S}_j^{WC}(\mathbf{x})$ is a set of feature indices that the magnitude of corresponding local attribution is smaller than $|\phi_j^{UFG}(\mathbf{x})|$. Practically, $\mathcal{M}_j^{WC}(x, n; N)$ can be equivalently derived by thresholding with $n/N$ quantile of absolute local attributions. To implement the smooth change of features, we gradually apply the mask with the scale factor proportional to the current step $n$. The sequential relation with WC mask is defined as,

$$\tilde{\mathbf{x}}(n+1) = \frac{n}{N}\mathcal{M}^{WC}(\tilde{\mathbf{x}}(n), n; N) \odot \tilde{\mathbf{x}}(0). \tag{8}$$

Figure 4 (a) depicts the sequence of distilled inputs $\tilde{\mathcal{X}}$ with WC mask. We can identify that the distillation by WC mask can remove the uninformative information (e.g., human body) to predict the object class, *French horn*.

However, we observe that considering only WC mask can be difficult to distill irrelevant features, if the strong local attribution is temporarily assigned to such features. For example, in Figure 4 (a), the human face (red box) remains until the end of the distillation with strong attribution. This phenomenon can be caused when the noise exists in the gradient or the value of feature itself is too large, which results in extremely high contributions for pixels without relevant information to predict the target class. Once such pixels are assigned with high attribution values, WC mask repeatably reassigns the masks to the same pixels and makes the overall distillation sequence be saturated. The saturated distillation sequence $\tilde{\mathcal{X}}$ disturbs the strength of multiple ablated inputs to build reliable attribution.

We define additional mask to reduce the saturation by filtering out features with extremely strong attribution. We call this mask as Extreme Positive Contributor (EPC) mask $\mathcal{M}^{EPC}(\cdot)$ and it is formulated as,

$$\mathbb{S}_j^{EPC}(\mathbf{x}) = \left\{ k \middle| \ \phi_k^{UFG}(\mathbf{x}) \leq \phi_j^{UFG}(\mathbf{x}) \right\} \tag{9}$$

$$\mathcal{M}_j^{EPC}(\mathbf{x}; q) = \begin{cases} 1 & \text{if } \frac{|\mathbb{S}_j^{EPC}(\mathbf{x})|}{dim(\mathbf{x})} \leq q \\ 0 & \text{otherwise .} \end{cases} \tag{10}$$

where $q$ is EPC threshold to control the ratio of ablation. Finally, we combine two masks for our distillation algorithm with relative weights w.r.t. the current step $n$ as,

Table 1: Comparison of various attribution methods with LeRF and MoRF on three models.

|  |  | G*I | GBP | IG | FG | GIG | DGA |
|---|---|---|---|---|---|---|---|
| LeRF (↑ is better) | VGG-16 | 0.078 | 0.113 | 0.096 | 0.415 | 0.110 | **0.434** |
|  | ResNet-18 | 0.114 | 0.145 | 0.158 | 0.448 | 0.185 | **0.533** |
|  | Inception-V3 | 0.171 | 0.162 | 0.243 | 0.558 | 0.255 | **0.691** |
| MoRF (↓ is better) | VGG-16 | 0.045 | 0.094 | 0.036 | 0.110 | 0.029 | **0.023** |
|  | ResNet-18 | 0.050 | 0.124 | 0.038 | 0.131 | 0.029 | **0.019** |
|  | Inception-V3 | 0.105 | 0.145 | 0.066 | 0.175 | 0.061 | **0.041** |

$$\tilde{\mathbf{x}}(n+1) = \left( \frac{n}{N} \mathcal{M}^{WC}(\tilde{\mathbf{x}}(n), n; N) + (1 - \frac{n}{N}) \mathcal{M}^{EPC}(\tilde{\mathbf{x}}(n); q) \right) \odot \tilde{\mathbf{x}}(0) \tag{11}$$

We note that in early distillation step, EPC mask takes high weight to reduce saturation at too highly attributed feature. In the late stage, WC mask gains high weight to remain the relevant features. The distillation sequence $\tilde{\mathcal{X}}$ with WC and EPC masks is shown in Figure 4 (b). We identify that using both WC and EPC masks removes irrelevant features (red box) and iteratively assign the attribution to relevant features (blue box).

### 3.3 Attribution Aggregation

With $N$ distillation steps, we obtain $N$ local attributions. The remaining question is how to aggregate these local attributions to acquire the final attribution. Likewise previous studies [Selvaraju *et al.*, 2017; Kindermans *et al.*, 2018; Bach *et al.*, 2015], we desire to take the positive contribution from each local attribution. Thus, we take the ReLU before the aggregation.

$$\phi^{DGA}(\mathbf{x}) = \frac{1}{N} \sum_{n=1}^{N} \text{ReLU}(\phi^{UFG}(\tilde{\mathbf{x}}(n))). \tag{12}$$

Finally, we call the unification of preceding modules as Distilled Gradient Aggregation (DGA) method. DGA method consists of the distillation algorithm with WC and EPC masks to generate the ablated inputs to achieve the local attribution, and the aggregation process considering features with the positive attributions. The illustration of the overall structure is depicted in Figure 1 and pseudo code is provided in Algorithm 1. To analyze the influence of each module (WC/EPC mask and ReLU), we perform the ablation study for each module and the results are available in Appendix A and H.

## 4 Experimental Results

In this section, we validate the effectiveness of DGA by both quantitative and qualitative comparison. We note that evaluating the attribution method is still challenging due to the absence of the ground truth. With this difficulty, we verify if the proposed method appropriately reflects the behavior of the model prediction by providing the quantitative comparison using three metrics: (1) pixel perturbation [Samek *et al.*, 2016], (2) sensitivity-$n$ [Ancona *et al.*, 2018] and (3) RemOve-And-Retrain (ROAR) [Hooker *et al.*, 2019]. Then we also provide the qualitative comparison among different attribution methods. In the following experiments, we set the hyperparameters for DGA as $N=30$ and $q=0.9$ with simple grid search. The details for the hyperparameter exploration is available in Appendix K. We select various gradient-based attribution methods as the baselines: Gradient*Input (G*I), Guided BackPropagation (GBP), Integrated Gradients (IG), FullGrad (FG), and GuidedIG (GIG).

### 4.1 Pixel Perturbation

Pixel perturbation is widely used method to benchmark the attribution methods if they correctly capture the relevance between the input features and the model output. To quantify the relevance

Table 2: Comparison of various attribution methods with sensitivity-$n$ on ResNet-18

| q% | 10 | 20 | 30 | 40 | 50 | 60 | 70 | 80 | 90 |
|---|---|---|---|---|---|---|---|---|---|
| G*I | -0.006 | -0.009 | -0.009 | -0.013 | -0.007 | -0.014 | -0.016 | -0.022 | -0.037 |
| GBP | 0.022 | 0.032 | 0.025 | 0.024 | 0.022 | 0.027 | 0.024 | 0.024 | 0.020 |
| IG | 0.013 | 0.022 | 0.020 | 0.027 | 0.032 | 0.037 | 0.039 | 0.047 | 0.067 |
| GIG | 0.006 | 0.007 | 0.004 | 0.002 | 0.003 | 0.002 | 0.002 | 0.002 | -0.001 |
| FG | -0.045 | -0.017 | -0.005 | 0.001 | 0.003 | 0.003 | 0.008 | 0.008 | 0.004 |
| **DGA** | **0.095** | **0.096** | **0.101** | **0.098** | **0.095** | **0.089** | **0.083** | **0.080** | **0.079** |

between the input features and the model output, pixel perturbation method removes the pixel values in order of relevance obtained by attribution methods. Then it measures the AUC of the softmax output change for the target class with the perturbation. There are two orders of removal, Most-Relevant-First (MoRF) to remove the pixels with top $k\%$ relevance and Least-Relevant-First (LeRF) to remove the pixels with bottom $k\%$ relevance. If input feature is actually highly related to the model prediction, the softmax output should decrease steeply when it is removed. Thus, MoRF is better if it is lower. In the same manner, higher LeRF is better.

We use 50k images of the validation set provided by ImageNet [Russakovsky *et al.*, 2015]. We use three publicly available pre-trained models: VGG-16 [Simonyan and Zisserman, 2015], Inception-v3 [Szegedy *et al.*, 2016], ResNet-18 [He *et al.*, 2016]. Table 1 indicates MoRF and LeRF results for the various attribution methods and model architectures. We identify that DGA shows the best performance in both MoRF and LeRF measure on entire architectures. We also provide additional comparison with different attribution methods (e.g., LRP) in Appendix I.

## 4.2 Sensitivity-$n$

Sensitivity-$n$ [Ancona *et al.*, 2018] has been suggested to quantify the generalization of previously suggested properties, Completeness [Sundararajan *et al.*, 2017] and Summation to Delta [Shrikumar *et al.*, 2017]. In general, since not all deep learning models can satisfy Sensitivity-$n$, we use empirical approximations to determine whether there is an algorithmic bias in the process of calculating attribution. To measure how much the attribution method satisfies this properties, the metric quantifies the correlation between the sum of the attributions ($\sum_{i \in S} \phi_i(\mathbf{x})$) for any subset of features ($S$), and the change of the model output when the input subset is ablated ($\mathbf{x}_{S^c}$). The metric can be represented as,

$$\text{corr}_S\left[\sum_{i \in S} \phi_i(\mathbf{x}), f(\mathbf{x}) - f(\mathbf{x}_{S^c})\right] \approx \text{corr}_{M_q}\left[\langle \Psi(M_q), \phi(\mathbf{x})\rangle, f(\mathbf{x}) - f\left((1 - \Psi(M_q)) \odot \mathbf{x}\right)\right] \quad (13)$$

where the subset $S$ is uniform randomly sampled with its cardinality $|S| = n$ and $S^C$ denotes its complement. The higher correlation value indicates that the attribution method empirically satisfies the sensitivity-$n$.

For computational efficiency, instead of ablating the entire individual input features ($n$), we perform patch-wise ablation to compute sensitivity-$n$ with randomly sampled binary masks $M_q \in \{0, 1\}^{14 \times 14}$ (100 masks for each input) where the portion of 1 in each mask is constrained be $q$. After the mask selection, we multiply the upsampled mask ($\Psi(M_q)$) to the attribution and the input to compute the correlation. The Table 2 shows the correlation values for various percentage of selection ($q$) over each attribution method for ResNet-18. We identify that DGA has the highest correlation values in entire cases.

## 4.3 RemOve-And-Retrain (ROAR)

ROAR is another metric to evaluate how well the attribution method captures the relevance of feature in the perspective of the model training. ROAR is performed by measuring the performance of the re-trained model with inputs modified according to relative ordering of the attribution. Each input in the dataset is modified by removing pixels with top $k\%$ attribution and replacing them with the average pixel value of the input. We perform ROAR experiment with simple CNN (6 Conv + 3

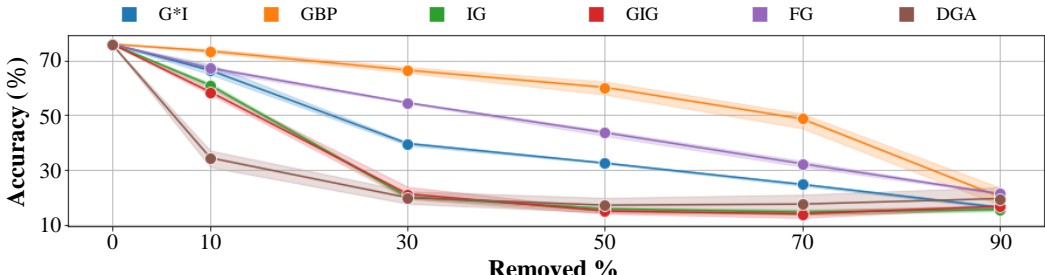

Figure 5: Comparison of ROAR experiment results on CIFAR-10 dataset among various attribution methods. The test accuracy for corresponding the percentage of removal.

Linear) trained on 50k images of training set provided by CIFAR-10 dataset [Krizhevsky, 2009] using Adam optimizer with learning rate 3e-4 and 100 epochs. After training, the performance of the model is quantified using the standard test dataset with 10k images. We note that the attribution method captures more relevant features if the test accuracy is lower. We provide the average performance over 10 trials for each attribution method, where the parameters are random initialized at each trial and fixed between attribution methods. Figure 5 shows the test accuracy measure in the ROAR experiment for each attribution method. The result indicates that the model trained on the modified dataset with DGA steeply decreases the test accuracy even with 10% removed. We conclude that DGA can extract the features which are relevant to training procedure in DNNs.

### 4.4 Qualitative comparison

We qualitatively compare the various attribution methods by visualizing the attribution heatmap and top 10% most relevant features at the same time. In Figure 6, we provide the result of randomly selected images from the validation set of ImageNet with the pre-trained VGG-16. We can identify that the attributions are more aligned with the object comparing other methods. For example, in the right-top row, DGA focuses the person who grabs a paddle while almost methods distribute the relevant pixels to sky and ocean. Although FG concentrates to the person, the relevant patch has less sharp than patch of DGA. We provide more examples and results for different models in Appendix B-D.

## 5 Discussion

In this paper, we propose a novel gradient-based attribution method, Distilled Gradient Aggregation (DGA). We provide the vulnerability of FG against the input perturbation and the counter-intuitive behavior of IG. To complement the weakness of both methods, we propose the gradient aggregation method along the distillation sequence that generates the inputs which the impurities are distilled. Our method obtains high quality attributions with its sharpness and object-alignment, and we verify the method through pixel perturbation, sensitivity-$n$, and ROAR evaluation metrics. We believe that our DGA method can be broadly applied to explain a decision of various DNNs.

**Broader Impact** Transparency of deep models is a matter of the highest priority for the application of such models in the real world, e.g., medical diagnosis [Caruana *et al.*, 2015] and autonomous driving [Yurtsever *et al.*, 2020]. We believe that providing the evidence which is well-aligned with the model decision would help the users of such applications to place great trust and the developers to improve or fix the model for better performance. Discovering unintended biases in the model is another issue [Stock and Cisse, 2018]. Such biases may occur from the dataset [Kim *et al.*, 2018] or the model itself. Identifying the root cause and removing such biases would be another expected future work, beyond the explanation on the input features.

**Limitation** Although our method has empirically outperformed in qualitative comparison and various quantitative experiments compared to previous work, the notion of better input attribution method is still vague. In this work, we adaptively find the sequence of inputs by using local attribution, but there would exist better justification of the sequence or the set of inputs that are essential clues for identifying the core features in the input.

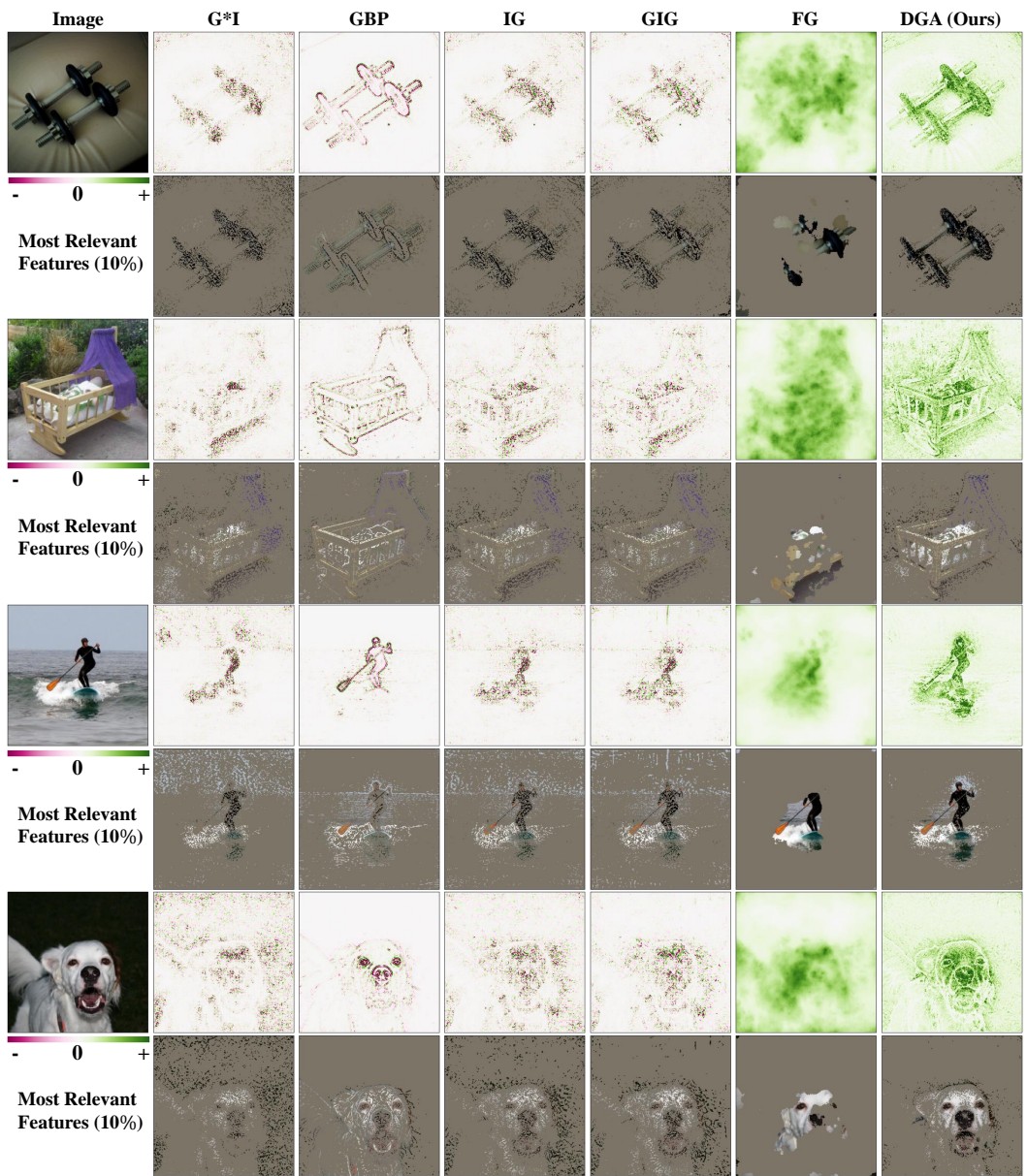

Figure 6: Qualitative comparison among various attribution methods for VGG-16 in the validation dataset of ImageNet. Upper rows describe the heatmaps obtained by each methods and lower rows shows top 10% most relevant input features. DGA generates sharp and object-oriented attribution heatmap in the almost examples. See more examples in Appendix B-D.

# Acknowledgement

This work was conducted by Center for Applied Research in Artificial Intelligence (CARAI) grant funded by Defense Acquisition Program Administration (DAPA) and Agency for Defense Development (ADD) (UD190031RD) and partly supported by Institute of Information & communications Technology Planning & Evaluation (IITP) grant funded by the Korea government (MSIT) (No.2022-0-00984, Artificial Intelligence, Explainability, Personalization, Plug and Play, Universal Explanation Platform), (No.2019-0-00075, Artificial Intelligence Graduate School Program (KAIST)), (No. 2022-0-00184, Development and Study of AI Technologies to Inexpensively Conform to Evolving Policy on Ethics).

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
