# A    Qualitative comparison for ablation study

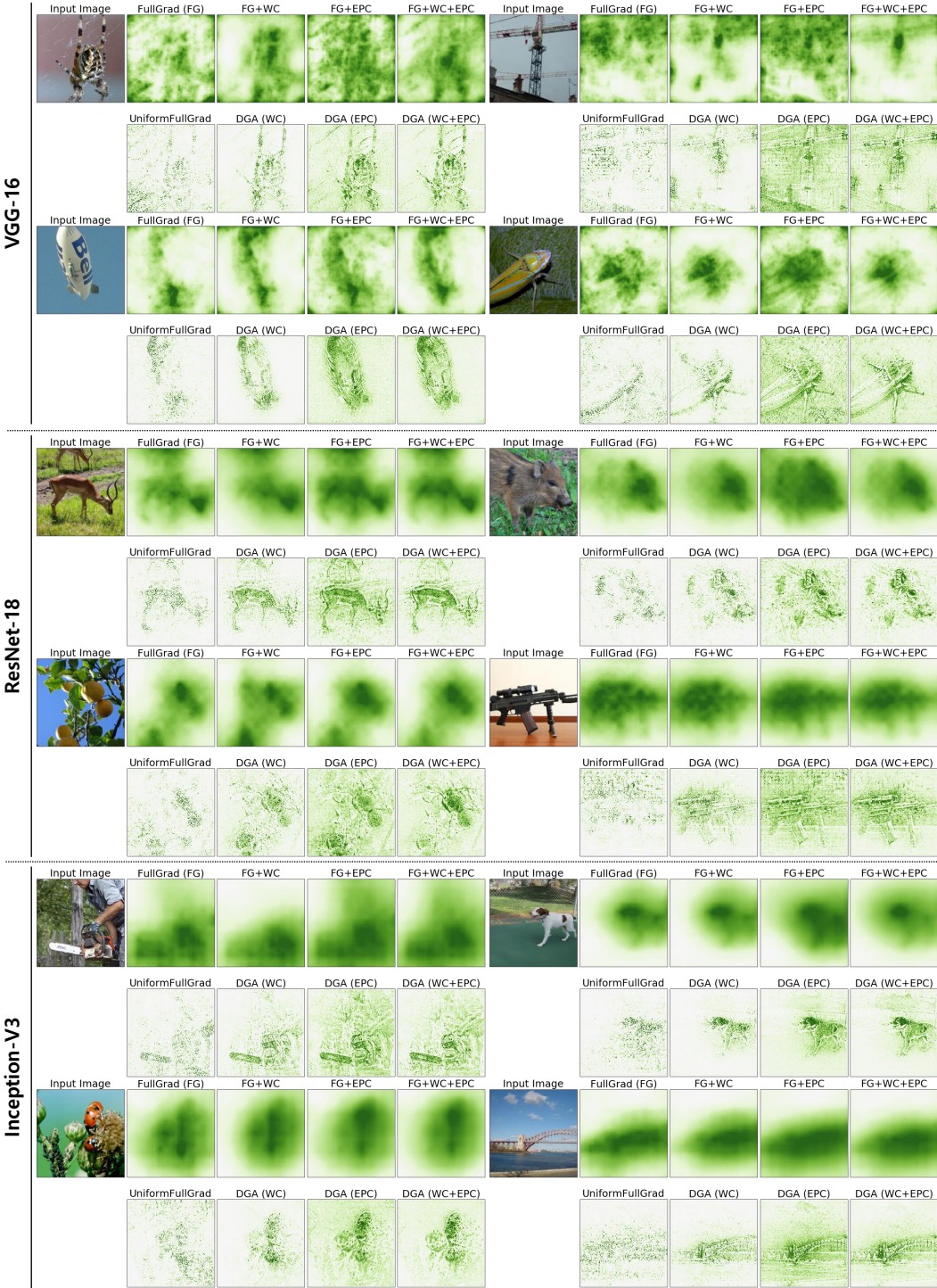

Figure 1: Qualitative comparison over the ablation of (1) FullGrad post-processing (UniformFullGrad), (2) WC mask (WC) and (3) EPC mask (EPC) on different model architectures. The results confirm that the post-processing helps to improve the resolution of the attribution. WC and EPC masks also help to concentrate on the object-aligned features.

# B Comparison between various attribution methods: VGG-16

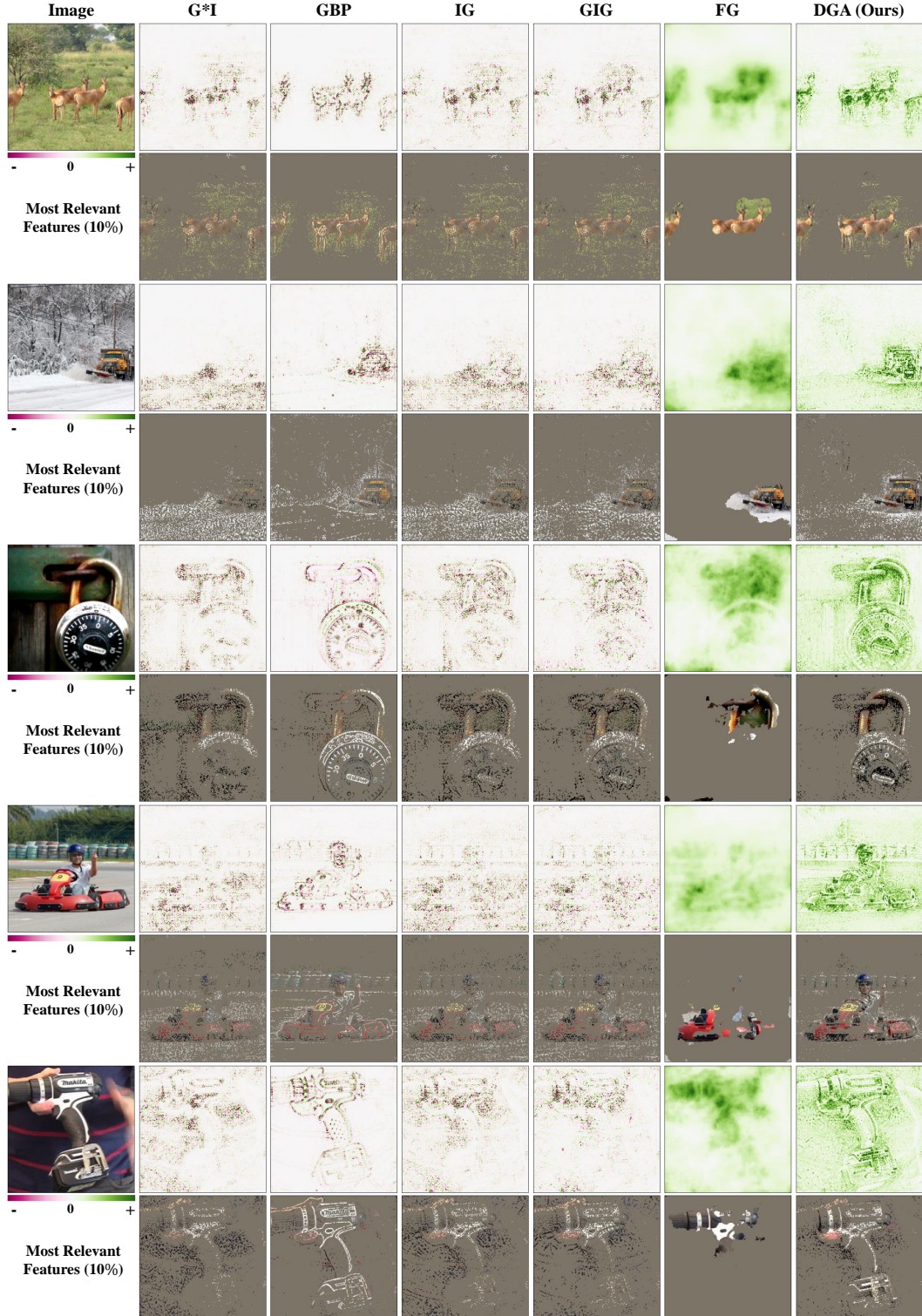

Figure 2: Qualitative comparison between various attribution methods in VGG-16. Odd rows depict the attributions and even rows depict the top pixels with top 10% attribution.

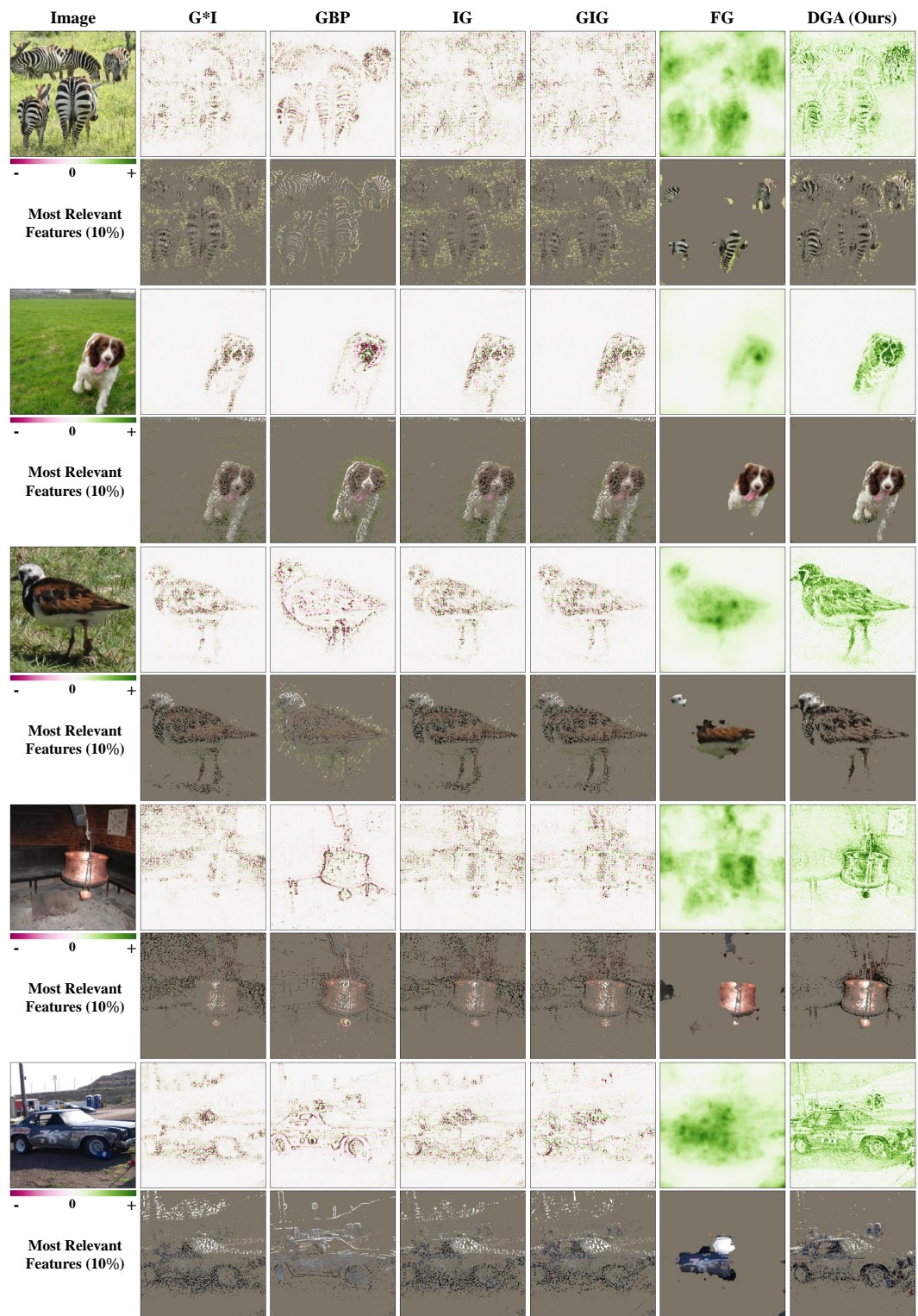

Figure 3: Qualitative comparison between various attribution methods in VGG-16. Odd rows depict the attributions and even rows depict the top pixels with top 10% attribution.

## C  Comparison between various attribution methods: ResNet-18

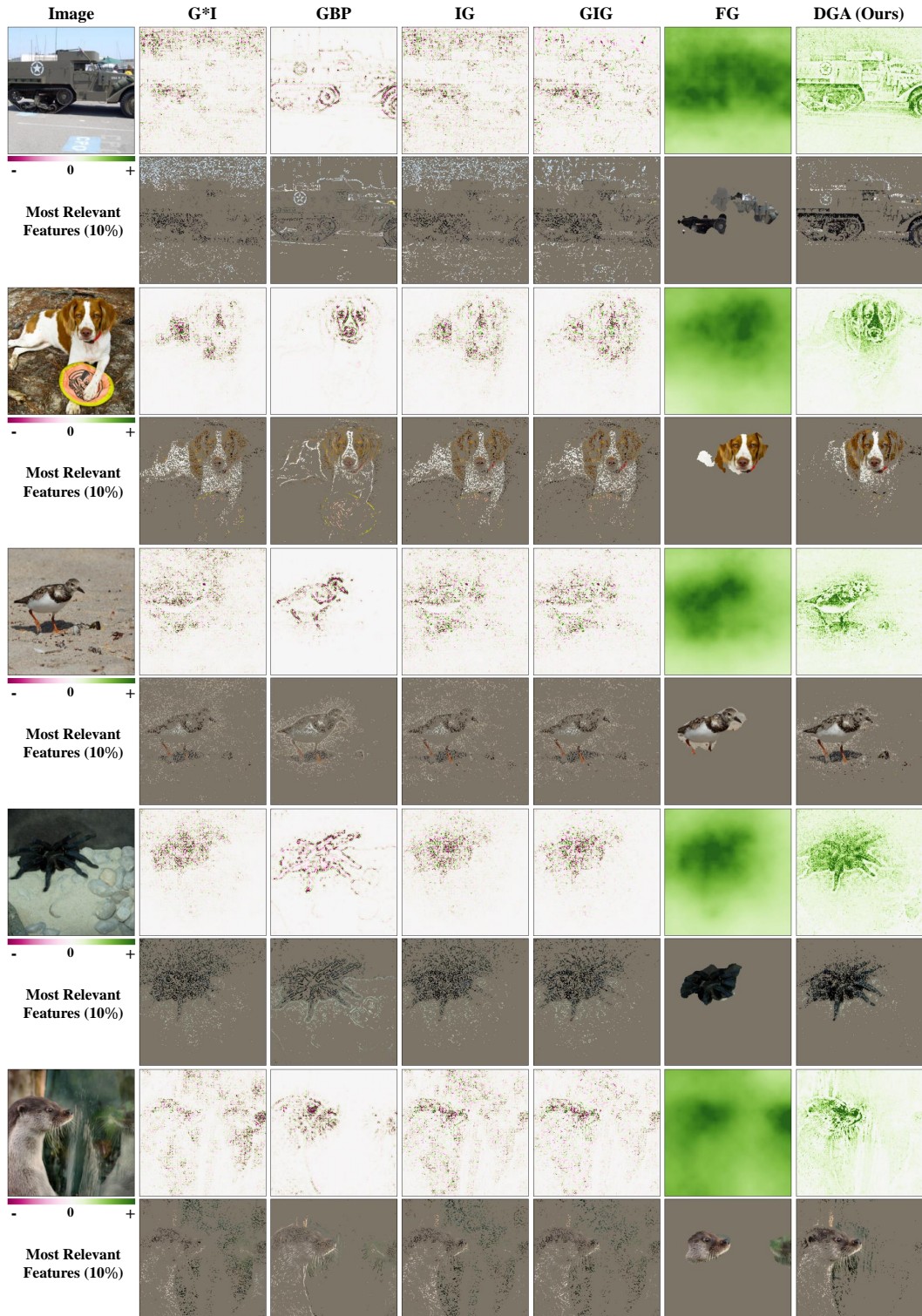

Figure 4: Qualitative comparison between various attribution methods in ResNet-18. Odd rows depict the attributions and even rows depict the top pixels with top 10% attribution.

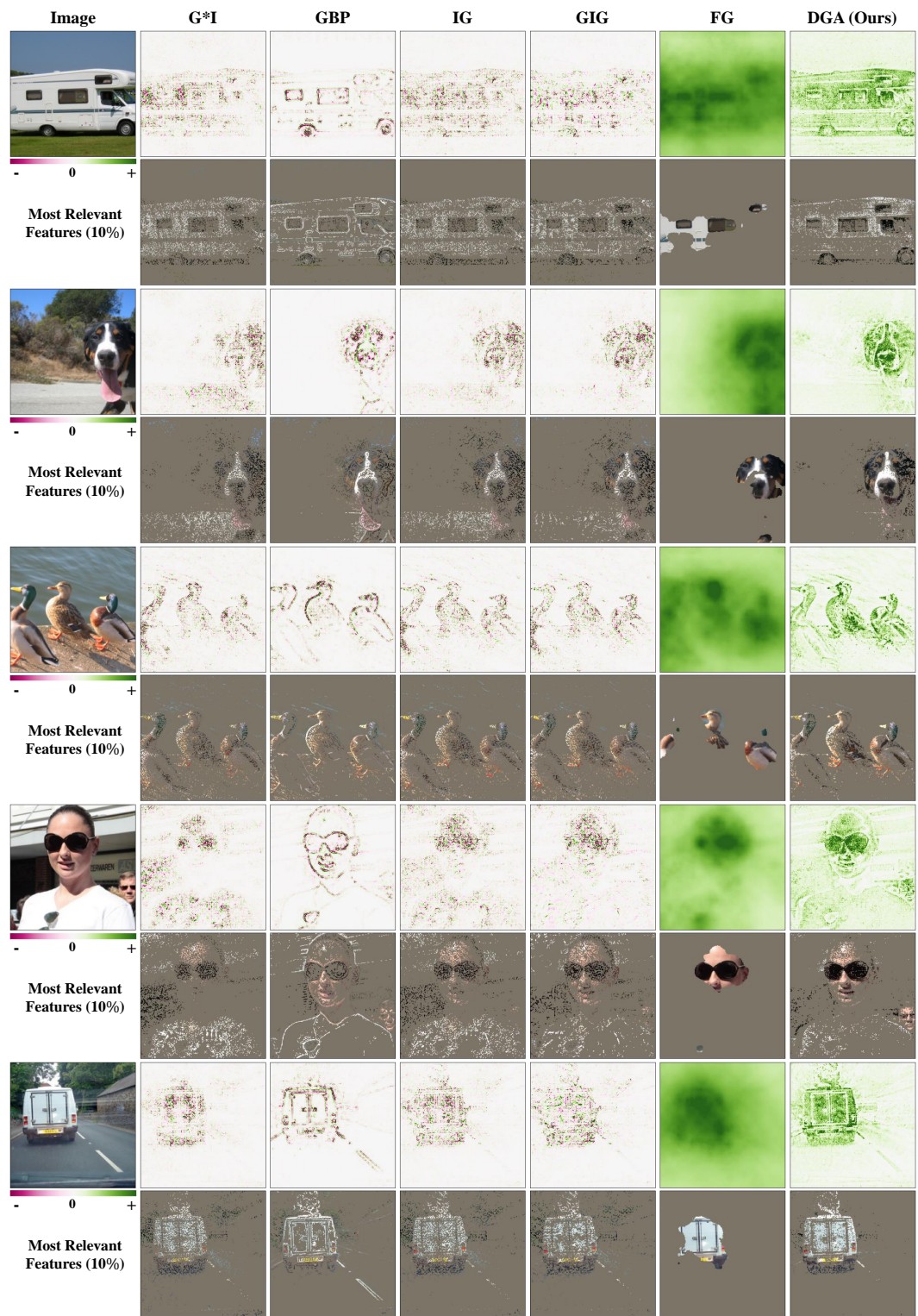

Figure 5: Qualitative comparison between various attribution methods in ResNet-18. Odd rows depict the attributions and even rows depict the top pixels with top 10% attribution.

# D    Comparison between various attribution methods: Inception-v3

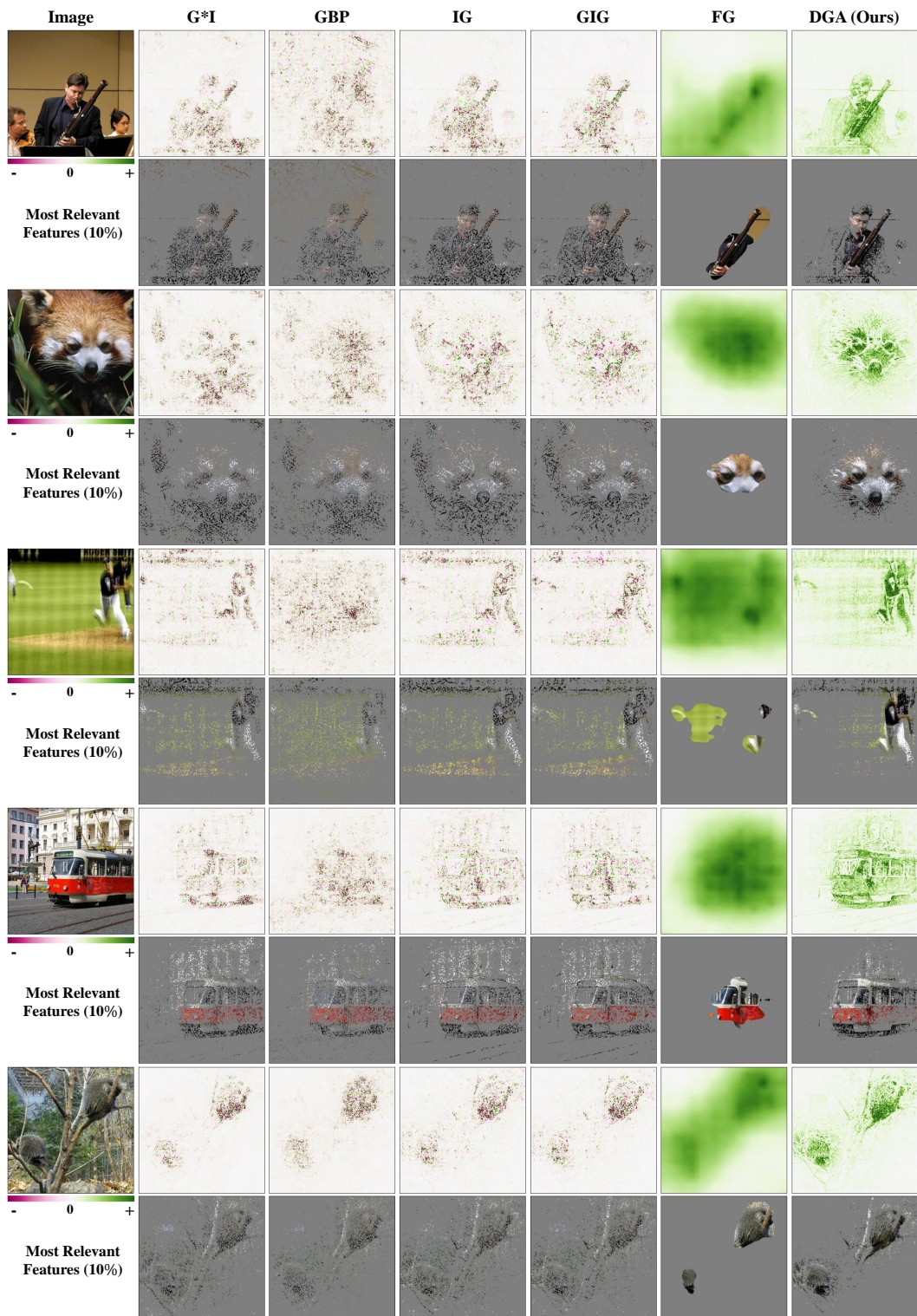

Figure 6: Qualitative comparison between various attribution methods in Inception-v3. Odd rows depict the attributions and even rows depict the top pixels with top 10% attribution.

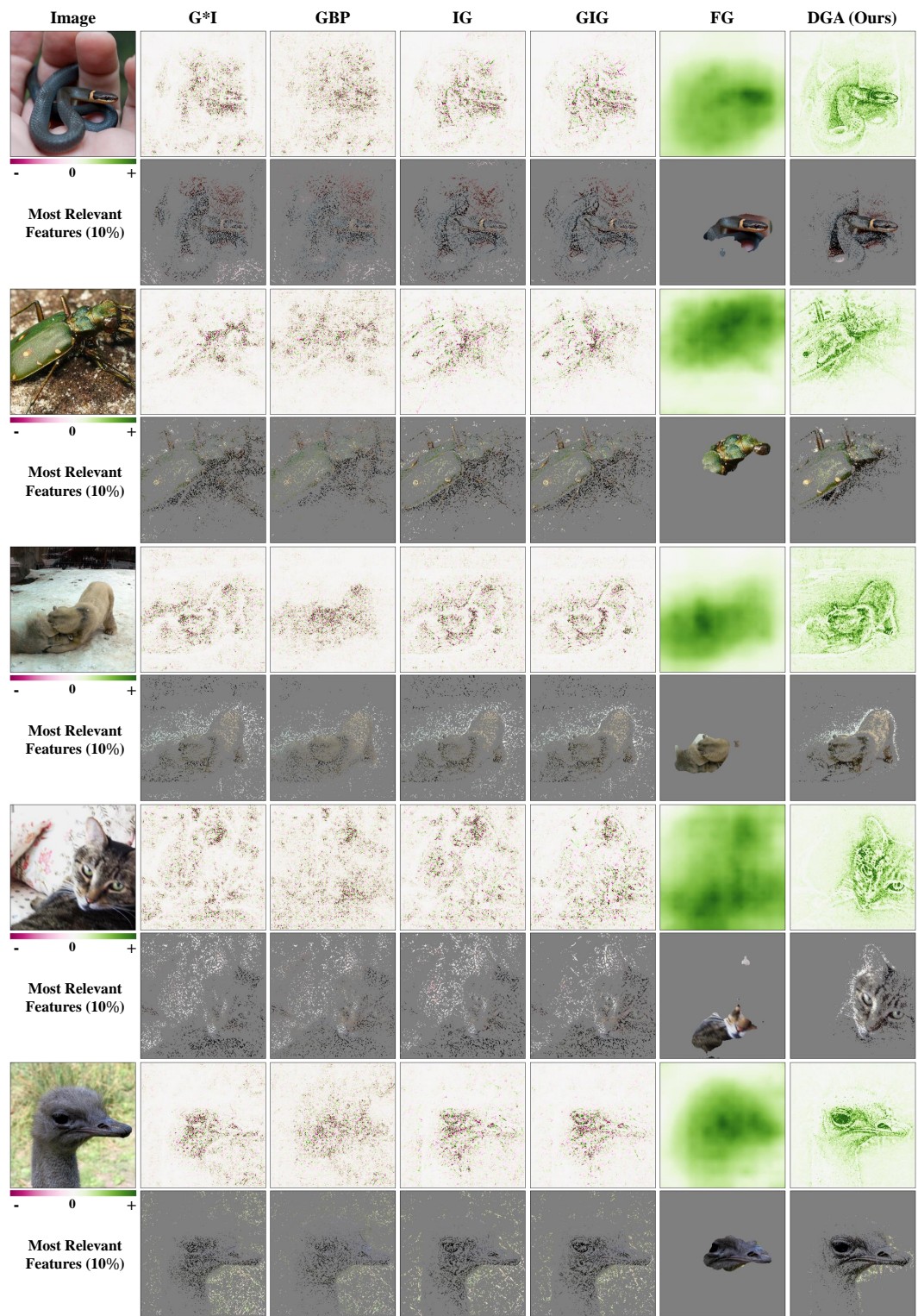

Figure 7: Qualitative comparison between various attribution methods in Inception-v3. Odd rows depict the attributions and even rows depict the top pixels with top 10% attribution.

 # E Comparison of the sequence generated by various methods in VGG-16

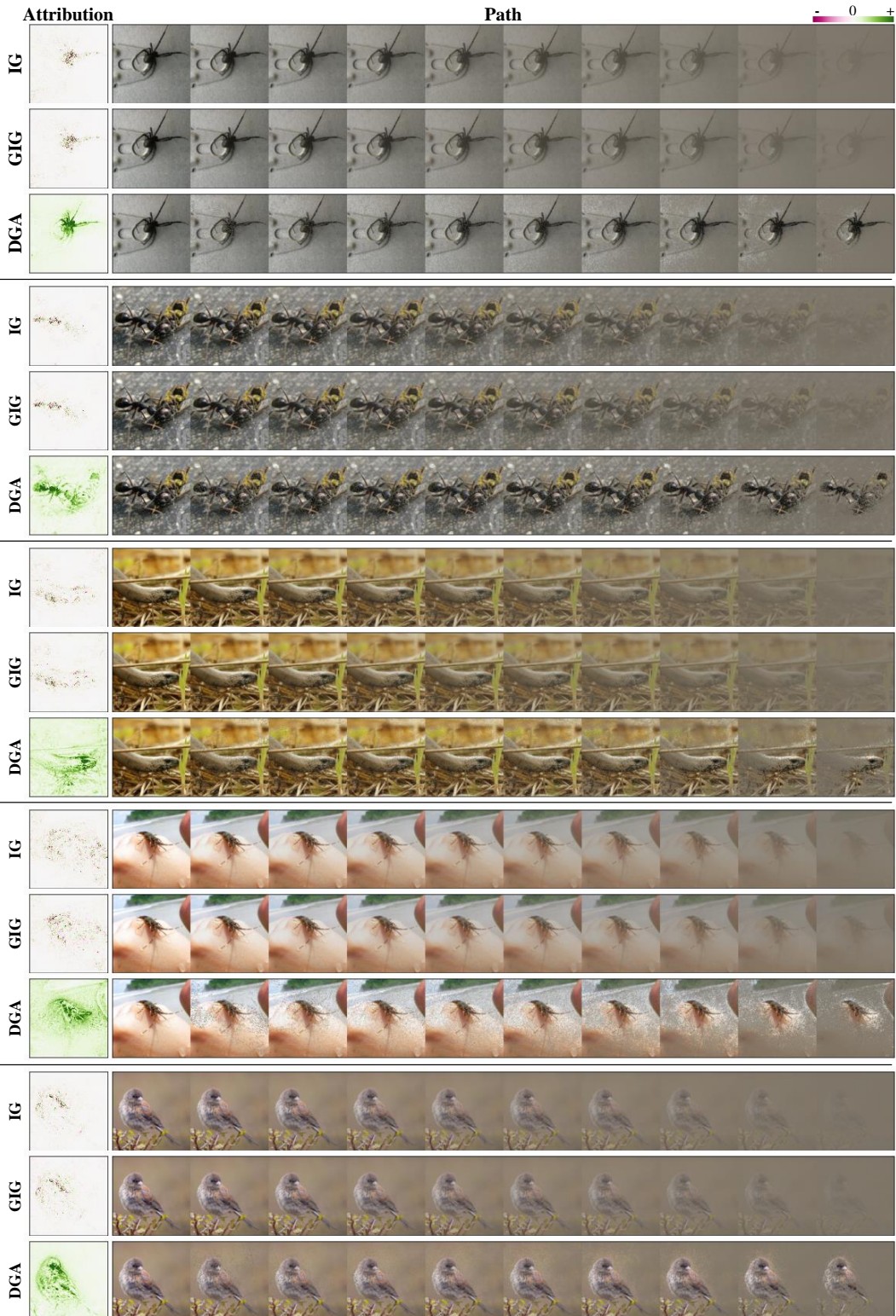

Figure 8: Comparison of utilized path for each input attribution method in VGG-16.

## F Implementation Code

We provide the simple implementation of our algorithm in Python language.

```python
import torch
def DGA(
    inputs,         # (tensor) Input tensor to be explained.
    model_func,     # (function) Model function to be explained.
    target,         # (int) Target class to be explained.
    epc_ratio=0.9,  # (float) Quantile threshold for EPC mask.
    n_steps=30,     # (int) Number of steps to aggregate attribution.
    ):

    attribution = 0
    cur_input = inputs.clone().requires_grad_()
    for _step in range(n_steps):
        # compute the local attribution at current input
        local_attr = local_attribution(cur_input, model_func=model_func, target=target)

        with torch.no_grad():
            # compute WC mask using low absolute quantile
            thres = local_attr.abs().flatten(1).quantile((_step+1)/n_steps, dim=-1, keepdims=True)
            thres = thres.unflatten(-1, [1,1,1])
            WC_mask = 1-(local_attr.abs() <= thres).float()

            # compute EPC mask using high positive quantile
            thres = local_attr.flatten(1).quantile(epc_ratio, dim=-1, keepdims=True)
            thres = thres.unflatten(-1, [1,1,1])
            EPC_mask = 1-(local_attr >= thres).float()

            # combine WC and EPC masks with the ratio depending on the current step
            final_mask = ((_step+1)/n_steps)*WC_mask + (1-(_step+1)/n_steps)*EPC_mask

            # take the positive attributions
            attribution += torch.relu(local_attr)

            # sample a new masked input using WC and EPC combined mask
            cur_input = torch.mul(inputs, final_mask)

    # normalize with the number of steps
    attribution = 1/n_steps * attribution
    return attribution
```

## G Pixel flip evaluation with different local attribution method in DGA

As DGA is computed by aggregating over the local attribution over the distillation sequence, replacing different local attribution method is applicable. With changing the local attribution method, it does not only change the attribution at each step, but also change the distillation sequence. Table 1 shows the pixel flip evaluation on Grad*Input (G*I), DGA using G*I, and DGA. The result indicates that the proposed DGA achieves the best performance in both LeRF and MoRF metrics. We also note that the proposed aggregation on the distillation sequence itself still induces the performance improvement, which can be verified by comparing G*I and DGA (G*I).

Table 1: Comparison of various attribution methods with LeRF and MoRF on three models.

|  | LeRF ($\uparrow$ is better) | | | MoRF ($\downarrow$ is better) | | |
|---|---|---|---|---|---|---|
|  | G*I | DGA (G*I) | DGA | G*I | DGA (G*I) | DGA |
| VGG-16 | 0.078 | 0.420 | **0.434** | 0.045 | 0.028 | **0.023** |
| ResNet-18 | 0.171 | 0.506 | **0.533** | 0.105 | 0.028 | **0.019** |
| Inception-V3 | 0.114 | 0.670 | **0.691** | 0.050 | 0.066 | **0.041** |

## H   Ablation study using pixel flip evaluation

We provide the ablation study on (1) the usage of ReLU and (2) WC/EPC masks in this section. To analyze the influence of ReLU, we compare IG and GIG with ReLU applied. We provide two variants of applying ReLU,

$$\text{IG}_{p1} = ReLU\Big(\int_{\alpha=0}^{1} \frac{dF(\gamma(\alpha))}{d\gamma_i(\alpha)} \frac{d\gamma_i(\alpha)}{d\alpha} d\alpha\Big), \tag{1}$$

$$\text{IG}_{p2} = \int_{\alpha=0}^{1} ReLU\Big(\frac{dF(\gamma(\alpha))}{d\gamma_i(\alpha)} \frac{d\gamma_i(\alpha)}{d\alpha}\Big) d\alpha. \tag{2}$$

Variants of GIG, $\text{GIG}_{p1}$ and $\text{GIG}_{p2}$, are also given as the same. Table 2 indicates that DGA shows better performance than such ReLU variants. Such result supports that the combination of distillation and ReLU achieves high performance, rather than ReLU itself.

We also provide the experiments on variants of DGA, with WC only ($\text{DGA}_W$), EPC only ($\text{DGA}_E$) and both (DGA). The result indicates that using WC show better performance in LeRF, but EPC better performs in MoRF. To achieve better performance in both metrics, we suggest to use both masks.

Table 2: Comparison of various attribution methods with LeRF and MoRF on three models.

|  |  | $\text{IG}_{p1}$ | $\text{IG}_{p2}$ | $\text{GIG}_{p1}$ | $\text{GIG}_{p2}$ | $\text{DGA}_W$ | $\text{DGA}_E$ | DGA |
|---|---|---|---|---|---|---|---|---|
| LeRF ($\uparrow$ is better) | VGG-16 | 0.141 | 0.262 | 0.157 | 0.325 | 0.417 | 0.349 | **0.434** |
|  | ResNet-18 | 0.204 | 0.339 | 0.239 | 0.401 | **0.551** | 0.479 | 0.533 |
|  | Inception-V3 | 0.233 | 0.587 | 0.239 | 0.610 | **0.719** | 0.646 | 0.691 |
| MoRF ($\downarrow$ is better) | VGG-16 | 0.036 | 0.035 | 0.030 | 0.036 | 0.039 | **0.018** | 0.023 |
|  | ResNet-18 | 0.041 | 0.039 | 0.032 | 0.036 | 0.034 | **0.014** | 0.019 |
|  | Inception-V3 | 0.125 | 0.070 | 0.120 | 0.069 | 0.082 | **0.029** | 0.041 |

## I   Additional pixel flip evaluations with different methods

We provide the quantitative evaluation on different attribution methods. In experiments, 5 additional methods are considered: DeepLIFT (DLIFT) [Shrikumar *et al.*, 2017], LRP [Bach *et al.*, 2015], SmoothGrad (SG) [Smilkov *et al.*, 2017], RISE [Petsiuk *et al.*, 2018] and Grad-CAM (GCAM) [Selvaraju *et al.*, 2017]. Table 3 indicates that DGA still shows the best performance.

Table 3: Comparison of various attribution methods with LeRF and MoRF on three models.

|  |  | DLIFT | LRP | SG | RISE | GCAM | DGA |
|---|---|---|---|---|---|---|---|
| LeRF ($\uparrow$ is better) | VGG-16 | 0.095 | 0.240 | 0.360 | 0.393 | 0.414 | **0.434** |
|  | ResNet-18 | 0.143 | 0.348 | 0.375 | 0.410 | 0.429 | **0.533** |
|  | Inception-V3 | 0.159 | - | 0.548 | 0.500 | 0.554 | **0.691** |
| MoRF ($\downarrow$ is better) | VGG-16 | 0.027 | 0.045 | 0.064 | 0.130 | 0.111 | **0.023** |
|  | ResNet-18 | 0.041 | 0.062 | 0.078 | 0.115 | 0.115 | **0.019** |
|  | Inception-V3 | 0.109 | - | 0.114 | 0.144 | 0.123 | **0.041** |

## J    Training setup of the simple model for analysis

For the simple analysis in the Section 3.1, we train a simple neural network with 2-dimensional half-moon shaped synthetic dataset. We train the simple neural network with fully-connected layers. The number of nodes in the hidden layers are 5-3-1 repectively and ReLU activation function is used for entire neurons without the output layer. The binary cross-entropy loss and Adam optimizer with learning rate 5e-4 are used to train for 5000 epochs.

| Name | Shape | Activation |
|---|---|---|
| Input | $N \times 2$ | - |
| Fully Connected 1 | $N \times 5$ | ReLU |
| Fully Connected 2 | $N \times 3$ | ReLU |
| Fully Connected 3 | $N \times 1$ | - |

## K    Hyperparameter exploration

DGA takes two hyperparameters: (1) the number of distillation steps $N$, and (2) the EPC threshold $q$. For the hyperparameter exploration, we use the pre-trained ResNet-18 and measure MoRF and LeRF score for the randomly selected 1k images from the training dataset of ImageNet, which is not inluded in the qualitative evalutaion in the main paper. At first, we analyze the relation between the distillation steps $N$ and EPC threshold $q$. We perform the grid search with the distillation steps $N = [1, 10, 20, 30, 50, 100]$ and the EPC threshold $q = [0.6, 0.7, 0.8, 0.9, 1.0]$. We note that the distillation step $N = 1$ is equivalent to the local attribution for original input image and EPC threshold $q = 1$ indicate the distillation with only WC mask. Figure 9 shows the relations between $N$ and $q$ with MoRF/LeRF score. The marker dots/lines indicate the specific distillation steps $N$ and EPC threshold $q$.

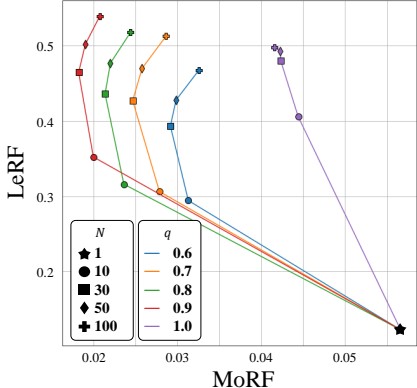

Figure 9: MoRF/LeRF score for various settings of the hyperparameters in ResNet-18.

From the left column in Figure 9, we observe that (1) the multiple steps (i.e., $N > 1$) always increase the performance of MoRF/LeRF regardless of the EPC threshold $q$, and (2) the low EPC threshold $q$ usually increase the performance of MoRF with degradation of the performance of LeRF. When we consider the importance of the MoRF/LeRF performance with same weight, the best choices of hyperparameters would be $N = 30, 50$ and $q = 0.9$. Although the $N = 50$ shows slightly better performance than $N = 30$, we select $N = 30$ for entire experiments in the main paper, because we benefit from the computational cost, which is directly related to $N$.