# OpenReview forum: "Distilled Gradient Aggregation: Purify Features for Input Attribution in the Deep Neural Network"
_NeurIPS.cc/2022/Conference — NeurIPS 2022 Accept_

### Official Review · Reviewer_5oaM · 2022-07-05

**Rating:** 3
**Confidence:** 5
**Soundness:** 1 poor
**Presentation:** 2 fair
**Contribution:** 2 fair

**Summary:**

This paper introduces a new explanation method called DGA for attributing a function’s output w.r.t its input features. DGA is motivated by problems of some related work, i.e. Integrated Gradient and FullGrad.

**Questions:**

My questions are discussed together with my comments in the weakness part. I will summarize them here and please refer to my comments above to see the relevant context.

- Why is the so-called “vulnerability” of FullGrad not a problem of the underlying model?

- What does “counter-intuitive behavior of IG” mean when an intuitive behavior is not discussed at first.

- How do the current metrics measure “vulnerability” and the so-called “counter-intuitive behavior”?

- Why RISE is missing from the evaluation section?


**Limitations:**

The authors have a separated section to discuss the limitations.

**Strengths And Weaknesses:**

### Strength
Visualizations are helpful to convey the authors’ message and explain the pipeline of the proposed method.

### Weakness

**Summary of Weakness.** My main concerns for this paper are problems in its motivation and the way the new method is evaluated. Although I find the current way the method is presented to the audience is quite confusing, this can be fixed in an easy way. I will vote for rejection for now.

**Motivations are not convincing.** A short summary for this paper’s motivation is the problems in existing methods, IG and FullGrad. Namely, FullGrad is vulnerable to local changes to the input (line 105)  and IG is counter-intuitive (line 114). However, none of the comments (or blames) made on the prior work makes sense to me. I will elaborate in detail.

- The fact that FullGrad (or gradient-based methods) may change for some $x’$ that is near to x does not mean this is a vulnerability of the explanation method. In fact, this is a problem of the underlying model and the explanation method is just being faithful to the model’s behavior. **In another word, being faithful is not a problem.** With $f(x) = f(x’)$ and $||x-x’|| \leq \epsilon$, it is unclear to me why $FullGrad(x)$ and $FullGrad(x’)$ must be the same if the model indeed treats $x$ and $x’$ differently in a significant way. In fact, a lot of work in adversarial vulnerability has shown that this is often the case: 1) linear regions are often tiny and very different from their neighboring regions [1]; and 2) models are learning noisy or spurious features, e.g. using background instead of the object, to make the prediction [2, 3, 4]. All these observations point out that making a more robust model is the way that one will observe that explanations for neighboring points look similar in the end [4, 5, 6]. If the underlying model does not behave similarly for similar input, why a **good** explanation method, e.g. FullGrad, should behave similarly? All in all, the so-called “vulnerability” is not a problem for FullGrad but a problem for the model.

- The comment about IG being counterintuitive is counterintuitive to me as the authors do not provide any justifications for what behavior is intuitive/expected. In the example shown in Figure 3c, the reason why IG is different in region A and B is that the baseline point is in A so that any integral path in A returns the same IG scores because the function is linear in A. In this case, the integral part of IG returns nothing but just the weight of the function, w^{(A)}, for every point in the path integral. Namely, $\forall x \in R^{(A)}, IG(x) = w^{(A)} \odot x$. And because you only move x on one axis, only the attribution score of IG for another axis will change. But for points in B, the baseline is outside the region B so IG(x) no longer just returns the weight $ w^{(B)}$: it is an aggregation of all $w^{(i)}$ the path integral traverse weighted by the length of the intersection between the path integral and the linear region. Thus, you observe IG also changes even when you only move the point on one axis. I have spent a long paragraph just showing that this is just a matter of baseline choice in IG and without any theoretical justifications or geometric-based analysis, I don’t see there is any problem in IG. Alternatively, if the authors argue that IG is using “bad” baseline points, then this observation should motivate the paper to find better baselines instead of developing DGA.

**Empirical evidence is not able to support the authors’ claims.** In the paper, the authors adopt metrics based on the ablation study using feature attributions to show that the proposed method, DGA, has a better performance on these metrics. Notice that the problems brought out earlier in this paper are **vulnerability** and **counterintuitive behavior** instead of something like “the current methods cannot identify features that are actually important to the model.” Without justifications about why the metrics used in the current paper, e.g. MoRF or ROAR, are able to measure **vulnerability** and *counterintuitive behavior**, the current empirical evidence is not related to the topic from my perspective.

**The way DGA is introduced is a bit confusing.** I find myself lost when transitioning from Section 3.1 to 3.2. I don't see clearly how RISE solves **vulnerability** and **counterintuitive behavior** in IG and FullGrad and I hope to see more principled justifications about that. Then the authors quickly start to explain the technical details of feature distillation, without a high-leveled discussion of what the method is going to do and why this approach solves the problem in a more efficient way compared to RISE. Also, since RISE also solves the problem as the authors mentioned in line 131, why is it missing from the evaluation section?


[1] Croce, Francesco et al. “Provable Robustness of ReLU networks via Maximization of Linear Regions.” AISTATS (2019).

[2] Ribeiro, Marco Tulio, Sameer Singh, and Carlos Guestrin. "" Why should i trust you?" Explaining the predictions of any classifier." Proceedings of the 22nd ACM SIGKDD international conference on knowledge discovery and data mining. 2016.

[3] Ilyas, Andrew, et al. "Adversarial examples are not bugs, they are features." Advances in neural information processing systems 32 (2019).

[4] Wang, Zifan et al. “Robust Models Are More Interpretable Because Attributions Look Normal.” (2021).

[5] Chen, Jiefeng, et al. "Robust attribution regularization." Advances in Neural Information Processing Systems 32 (2019).

[6] Singh, Mayank Kumar et al. “On the Benefits of Attributional Robustness.” ArXiv abs/1911.13073 (2019): n. pag.

---

> ### Author Response · Authors · 2022-08-02
> **Official response to Reviewer 5oaM (3/3)**
>
> ### [Q5] **Why RISE is missing from the evaluation section?**
>
> We comprise the comparative group of attribution methods mainly focused on the gradient-based methods, where RISE is a perturbation-based method, which does not use gradients. However, as reviewer pointed out, it would be necessary to compare the performance of RISE. Below table provides the pixel flip evaluation results to compare RISE and various attribution methods, where DGA still shows highest performance.
>
> We note that RISE shows prominent performance in LeRF, which we think the improvement came from the exploration of the multiple linear regions and the aggregation of them.
>
> This result is also attached to the updated experiments in the manuscript. We apologize for missing this evaluation and appreciate the suggestion.
>
> ### LeRF (Higher $\uparrow$ is better)
>
> $\begin{array}{l|cccccccc}
>  & \text{G\*I} & \text{GBP} & \text{IG} & \text{FG} & \text{GIG} &\text{RISE} & \text{DGA} \newline\hline
> \text{VGG-16} & 0.078 & 0.113 & 0.096 & 0.415 & 0.110 & 0.393 & \textbf{0.434} \newline
> \text{ResNet-18} & 0.114 & 0.145 & 0.158 & 0.448 & 0.185 & 0.410 &  \textbf{0.533} \newline
> \text{Inception-V3} &  0.171 & 0.162 & 0.243 & 0.558 & 0.255 & 0.500 & \textbf{0.691}
> \end{array}$
>
> ### MoRF (Lower $\downarrow$ is better)
>
> $\begin{array}{l|cccccccc}
>  & \text{G\*I} & \text{GBP} & \text{IG} & \text{FG} & \text{GIG} &\text{RISE} & \text{DGA} \newline\hline
> \text{VGG-16} & 0.045 & 0.094 & 0.036 & 0.110 & 0.029 & 0.130 & \textbf{0.023} \newline
> \text{ResNet-18} & 0.050 & 0.124 & 0.038 & 0.131 & 0.029 & 0.115 &  \textbf{0.019} \newline
> \text{Inception-V3} & 0.105 & 0.145 & 0.066 & 0.175 & 0.061 & 0.144 & \textbf{0.041}
> \end{array}$

---

> ### Author Response · Authors · 2022-08-02
> **Official response to Reviewer 5oaM (2/3)**
>
> ### [Q3] **How do the current metrics measure “vulnerability” and the so-called “counter-intuitive behavior”?**
>
> We think that the (1) vulnerability of the local attribution method and (2) counter-intuitive of the global attribution method sometimes fail to achieve the reliability in measuring the relative importance for the input features. It denotes that the computed attribution heatmap cannot interpret the prediction of the network exactly. In a perspective of the reliable explanation, we selected two metrics: (1) ROAR, and (2) Pixel filp (LeRF and MoRF) which can quantify the relation between ablation of input and the decision of the network.
>
> We additionally provide the evaluation result of sensitivity-$n$ [Ancona, et al., 2018] which quantify the satisfaction of the completeness (one axiom for the reliable attribution method). Please refer the answer to reviewer cvDy for the detailed evaluation procedure.
>
> According to the below table, we can identify that DGA has the highest correlation values among various attribution methods. This result also indirectly supports that DGA can resolve (1) the vulnerability of the local attribution, and (2) counter intuitive behavior of the global attribution to generate the reliable explanation.
>
> $\begin{array}{r|ccccccccc}
> q\\% & 10 & 20 & 30 & 40 & 50 & 60 & 70 & 80 & 90 \newline\hline
> \text{G\*I} & -0.006 & -0.009 & -0.009 & -0.013 & -0.007 & -0.014 & -0.016 & -0.022 & -0.037 \newline
> \text{GBP} & 0.022 & 0.032 & 0.025 & 0.024 & 0.022 & 0.027 & 0.024 & 0.024 & 0.020 \newline
> \text{IG} & 0.013 & 0.022 & 0.020 & 0.027 & 0.032 & 0.037 & 0.039 & 0.047 & 0.067 \newline
> \text{GIG} & 0.006 & 0.007 & 0.004 & 0.002 & 0.003 & 0.002 & 0.002 & 0.002 & -0.001 \newline
> \text{FG} & -0.045 & -0.017 & -0.005 & 0.001 & 0.003 & 0.003 & 0.008 & 0.008 & 0.004 \newline
> \text{DGA} & \textbf{0.095} & \textbf{0.096} & \textbf{0.101} & \textbf{0.098} & \textbf{0.095} & \textbf{0.089} & \textbf{0.083} & \textbf{0.080} & \textbf{0.079}
> \end{array}$
>
> [Ancona, et al., 2018] Ancona, Marco, et al. "Towards better understanding of gradient-based attribution methods for Deep Neural Networks." *International Conference on Learning Representations*. 2018.
>
> ---
>
> ### [Q4] **The way DGA is introduced is a bit confusing**
>
> We are sorry for the confusing manuscript. We are revising the manuscript to clarify the motivation for distillation process inspired by RISE.
>
> Based on the observations in Figure 2 and 3 of the main paper, we raise the weakness of the local and the global attribution methods (vulnerability and counter-intuitive behavior). To alleviate the weakness, we desire an attribution method to combine the strengths of each type of attribution methods.
>
> To resolve above desire step by step, we start from the local attribution which is free from the baseline selection. Then the remained problem is how to select the meaningful linear regions to generate the reliable attributions to interpret the decision of the network.
>
> For the strategy to select linear regions, RISE suggests the random perturbation-based approach. It explores the multiple linear regions with randomly ablated masks to improve the reliability of the attributions. However, randomized ablation requires the expensive cost to guarantee the exploration of linear regions which can reflect the global behavior of the network.
>
> We are also inspired from the adaptive exploration for the perturbed inputs in GIG [Kapishnikov, et al., 2021], which can improve the final attribution. Thus, we hypothesize that the adaptive exploration of linear regions based on the intermediate local attribution can reduce the cost of randomized exploration and follow the spirit of the adaptive exploration.
>
> Finally, we propose the sequential feature distillation algorithm to obtain a sequence of ablated inputs, and each perturbed input will reside each linear region. The distillation strategy proposed in the main paper is introduced as  WC and EPC masks.
>
> [Kapishnikov, et al., 2021] Kapishnikov, Andrei, et al. "Guided integrated gradients: An adaptive path method for removing noise." *Proceedings of the IEEE/CVF conference on computer vision and pattern recognition*. 2021.

---

> ### Author Response · Authors · 2022-08-02
> **Official response to Reviewer 5oaM (1/3)**
>
> We sincerely appreciate your time and efforts in reviewing our paper, as well as the constructive comments. We respond to each of your comments one by one. All the responses will be carefully incorporated in the final draft.
>
> ---
>
> ### [**Q1] Why is the so-called “vulnerability” of FullGrad not a problem of the underlying model?**
>
> We think that it is non-trivial to determine whether the change in the computed relative importance for input features in $\|\textbf{x}-\textbf{x}’\|\leq \epsilon$ with $f(\textbf{x})\approx f(\textbf{x}’)$ is caused by (1) the problem of the model or (2) the vulnerability of the attribution method.
>
> If the change of the attribution heatmap is determined by the behavior of the model, not the problem of the attribution method, we believe that the computed relative importance of the features will show high performance in LeRF and MoRF. Because LeRF and MoRF measure the model response according to the ablation of input features, such causal relationships would be reflected in the evaluation process.
>
> However, in Figure 2 of the main paper, we observe that FG shows drastic change of score in both LeRF and MoRF compared to other methods (IG or DGA). It denotes that the relative importance of input features computed by FG may not capture the reliable explanation to interpret the relationship between the perturbed inputs and the decision of the network.
>
> We conjecture that this phenomenon is caused by the local attribution which only considers the single linear region in the input space. On the other hand, such phenomenon is rarely observed in the global attribution methods (e.g., IG) which considers the multiple linear regions.
>
> As a result, we believe that the change of attribution for the small perturbations would be caused by the vulnerability of the local attribution method (e.g., FG), not the problem of the model.
>
> ---
>
> ### [**Q2] What does “counter-intuitive behavior of IG” mean when an intuitive behavior is not discussed at first.**
>
> We are sorry for the confusing descriptions in the main paper. We are revising the manuscript to clarify what is the counter-intuitive behavior of IG.
>
> We note that “counter-intuitive behavior of IG” is raised to describe the observation that the attribution computed by IG for input feature $x_2$ is changed when only the input feature $x_1$ is modified, which is represented in Figure 3 of the main paper.
>
> As the reviewer pointed out, the counter-intuitive behavior of IG is caused by the choice of the baseline. Because the choice of the baseline determines the integration path of IG, **the baseline selection can be regarded as determining** **(1) which linear regions are traversed by the path, and (2) how much portion of the path is included in each selected linear region.** However, it is non-trivial to select proper linear regions and adjust the weight of each linear region with only changing the baseline.
>
> Instead of finding the good baseline, we consider the exploration of a sequence of linear regions to generate reliable attribution to interpret the decision of the network. To this end, we propose the distillation process, which a select sequence of linear regions based on the intermediate local attribution.
>
> For the detailed discussion, please refer to our common response.

---

> ### Comment · Reviewer_5oaM · 2022-08-09
> **Response to Rebuttal**
>
> Sorry for my late response. I appreciate the updated version of the submission, which helps to explain the motivations of the work. I extremely appreciate the experiment of sensitivity-n and hope that is included in the revision.
>
> **The response does not address my concerns about the motivations.**
> -  For the response on the vulnerability on FullGrad, the authors claim that experiments on LeRF or MoRF show that FullGrad is not faithful enough to the model. I believe to discuss the faithfulness of explanations, one can not overlook within what neighborhood we are interested in capturing the model's behavior. That is, the faithfulness of an explanation is not well-defined without specifying the neighborhood. Some paper call this **distributional faithfulness** [1] or **infidelity** [2]. Back to the rebuttal, the ablations studies conducted on these metrics, e.g. LeRF, MoRF or sensitivity-N, usually ablates the features with zeros (or random noises). That is, it evaluates if the attribution methods captures the behavior of the model within a neighborhood "connecting" the original point (or the random noise) to the input of interest. Let's call this neighborhood as $\mathcal{A}$. However, FullGrad (or gradient itself) is the weight of first-order Taylor expansion of the function, which perfectly approximate the target function within the current linear region, if the target network is piece-wise linear, i.e. ReLU networks and let's call this linear region as $\mathcal{B}$. **My point is, showing FullGrad is not a faithful explanation in $\mathcal{A}$ does not serve as an evidence that FullGrad is not a faithful explanation in $\mathcal{B}$.** (In fact, [2] has shown the way to achieve the most faithful explanation given a particular neighborhood and how the most faithful explanation is related to completeness-preserving methods, e.g. IG.) This should explain the authors' conjecture *"We conjecture that this phenomenon is caused by the local attribution which only considers the single linear region in the input space. On the other hand, such phenomenon is rarely observed in the global attribution methods (e.g., IG) which considers the multiple linear regions."* With this being said, the linear approximation, e.g. FullGrad, is different between one linear region to another simpiliy means **the underlying model has very different local linear regions** for some small $\epsilon$-ball.
>
> - For the counter-intuitive behavior of IG, I am not sure the motivating example, i.g. the contribution of $x_1$ must remain if only $x_2$ changes, makes sense. This is because features can interact with others in a non-linear way so the change to $x_1$ may impact the contribution of $x_2$. Nevertheless, I think the authors are looking for some attribution function for $x_1$ that is provably not a function of $x_2$. In this case, a gradient integration with a linear path, i.e. IG, can't guarantee at all. But I am not sure by to what extend DGA is better because all metrics evaluated in paper are just indirect results. **Towards this end, an experiment shows that, when changing $x_1$ while fixing $x_2$ the contribution of $x_2$ calculated by DGA does not change while the contribution calculated by IG changes, seems to be the direct evidence to support the authors' claim that DGA solves the issue of IG.** Alternatively, any theoretical analysis shows DGA is better on this axis is also okay to me.
>
> I thank for all new results and I agree with other reviewers and the authors that all experimental results including ablation studies show that DGA is a better method than others: DGA is faithful in the neighborhoods the corresponding ablation methods create. This is mostly because DGA propose some nice local linear regions and the aggregation of gradients in these regions better approximate the model in a larger neighborhood. However, these are not evidences to support the claims the authors made about 1) vulnerability of gradients and 2) IG. The community who discusses the vulnerability of explanations usually run adversarial attacks on explanations, e.g [3]. For IG, I have proposed a small experiment above but it would be nicer to design a more convincing experiment.
>
> **In summary, my score remains because 1) the problems authors find are not convincing to be flaws of existing work to my perspective; and 2) experiments with good numbers, though following standard metrics frequently-used in the literature, are very indirect.**
>
> [1] Leino, Klas et al. “Influence-Directed Explanations for Deep Convolutional Networks.” 2018 IEEE International Test Conference (ITC) (2018): 1-8.
>
> [2] Yeh, Chih-Kuan et al. “On the (In)fidelity and Sensitivity of Explanations.” NeurIPS (2019).
>
> [3] Amirata Ghorbani, Abubakar Abid, and James Zou. 2019. Interpretation of neural networks is fragile. In Proceedings of the Thirty-Third AAAI Conference on Artificial Intelligence and Thirty-First Innovative Applications of Artificial Intelligence Conference

---

### Official Review · Reviewer_H2WE · 2022-07-07

**Rating:** 6
**Confidence:** 4
**Soundness:** 2 fair
**Presentation:** 3 good
**Contribution:** 3 good

**Summary:**

The paper identifies pitfalls in two attributions methods, then proposes a novel attribution method that purports to solve the problem.
1) The paper identifies issues with popular attribution methods Integrated Gradients and FullGrad. IG suffers from attribution noisiness and unintuitive results, while FullGrad has weak dependency.
2) The paper proposes using FullGrad with a different post-processing function, and engaging in sequential feature distillation. Essentially they calculate a FullGrad attribution; then they mask a constant, small proportion of the top-ranked and an increasing proportion of the bottom-ranked inputs according to the attributions. They repeat this process until the image is fully masked.
3) They run comparative tests on other attribution methods. They run quantitate ROAR and Perturbation tests, and they run qualitative tests. These tests indicate the advantages of their method.

**Questions:**

Q: Have you addressed the issue of not knowing the ground truth? How do you know this method is reliable if the ground truth is unknown? Pixel Perturbation and ROAR tests address this question, but perhaps another test that could help. Are there any instances where the ground truth is known, i.e., we know what part of the image contributes to the model classification?

Q: It seems your method consists of 1) using successive masking and reattributing, and 2) using Fullgrad to get attributions between iterations. Why use FullGrad and not any other attribution method for part 2? Have you tried your algorithm with other attribution methods? I wonder if the real novelty and success here is in part 1, your masking approach, and using FullGrad between iterations is replaceable.

Q: What made you choose the methods you used for comparison? Why not SmoothGrad, LRP, Deconvolutional Networks, DeepLift, GradCam, or CAMERAS (see references)? I am not saying you should use them all, and understand it would require more work to compare them all. I am wondering how you justify the comparisons you choose to make and which you leave out. Perhaps a blurb on why you choose to compare some and not others?

Q: Which baseline do you use for IG? It seems you are not using black or white baselines, since IG attributes to the dumbbells and white cloth in the first and second images of figure 2, and IG gives zero attribution to pixels with the same value as the baseline. Have you considered trying other baselines, as in [Pascal Sturmfels, 2020]? It is possible a single baseline IG is not the strongest formulation. If this is the case, It would be interesting to see what the comparative metrics are to a stronger version of IG that uses a better baseline. It would strengthen the claim that this method outperforms IG.

Q: It seems that the counter-intuitive behavior of IG is due to the choice of baseline. An unanswered question is: why choose that baseline? It makes sense to put it in the middle, between the two curves. However, another side-effect of IG is that if you leave the baseline at (0,0), then any point of the form (0,x_2) will have 0 attribution in the first component, which does not seem correct either. This is not a problem unique to this paper: the issue of how to choose a baseline is a generally unanswered for baseline-attribution methods. It has to do with the idea of missingness in game-theory, which is not present in object recognition. See references for treatment of baseline options.

Q Are there any references needed for section 3.1, "Vulnerability of FullGrad" and "Counter-intuitive behavior of IG"? These observations are completely original?

References for the Author:
- DeepLift
Avanti Shrikumar, Peyton Greenside, and Anshul Kundaje.
Learning important features through
propagating activation differences.
- Deconvolution Networks
Matthew D Zeiler and Rob Fergus. Visualizing and understanding convolutional networks.
- CAMERAS
Mohammad AAK Jalwana, Naveed Akhtar, Mohammed Bennamoun, and Ajmal Mian. Cameras:
Enhanced resolution and sanity preserving class activation mapping for image saliency.
- An investigation of various baselines used for IG.
Pascal Sturmfels, Scott Lundberg, S.-I. L. Vi-
sualizing the impact of feature attribution base-
lines, 2020.
- Using a distribution of training-set images as a baseline for IG.
Erion, G., Janizek, J. D., Sturmfels, P., Lundberg, S. M.,
and Lee, S.-I. Improving performance of deep learning
models with axiomatic attribution priors and expected
gradients.
- Uses a Gaussian blur process to alter the path of IG.
Xu, S., Venugopalan, S., and Sundararajan, M. Attribution


Small issues/typos:
- Autonomous Driving (22-23)
- "so that humans" (24)
- no "the" before "critical (26)
- "evidence for the "decisions" (27)
- a heatmap (30)
- What is Clever Hans (32)
- "Obtaining a trustworthy input" (32)
- Missing an arrow from "Input x" down to the first dot in Figure 1 (A)
- derives the attribution to be vulnerable (108)
- let we have two inputs (108-109)
- Figure 2 sentence 2 is incomplete.
- Please describe MoRF and LeRF in caption of Figure 2.
- What is x-bar in 121-12? I believe it is the baseline but I don't see in defined before.
-  each ablated features (133)
- that masking out irrelevant features using IG(138)
- as uniformly distributing function (142-143)
- to remain the relevant features (175)
- In Figure 2 b, I assume you pruned k%. What is k?
- Similarly, in Table 1, what is k? Can you give multiple k's in a graph? This is a snapshot using one k, and could not be representative.
- For pixel perturbations, did you calculate attribution rankings for each of 50k images for each method and model, then clip inputs according to the rankings and re-evaluate the confidence? If so, does table 1 report average new confidence, over all 50k images, after k% removed?
- while almost methods (234)
- concentrates to the person (235)
- the relevant patch has less sharp than patch of DGA (235-236)
- in the Appendix (236)
- we provide the vulnerability (239)
- rework sentence on line 241
 - there would exist... (256)

**Limitations:**

Some limitations mentioned. Other limitations could be addressed refereeing to Questions section.

**Strengths And Weaknesses:**

Strengths:
- Clear presentation, easy to follow with good examples.
- Strong qualitative and ROAR results support the conclusion.
- Masking and reattributing successively seems novel, to my understanding.

Weaknesses:
- Motivation seems weak. Why did the authors identify those two issues in two seemingly arbitrary methods as motivation for their method? Why did they decide to use FullGrad with the successive masking, as opposed to any other method?
- Grammar/typos need polish.
- Some experimental choices seem arbitrary/unexplained (see questions).

---

> ### Author Response · Authors · 2022-08-02
> **Official response to Reviewer H2WE (3/3)**
>
> ### [Q5] Counter-intuitive behavior comes from the baseline selection
>
> As the reviewer pointed out, the counter-intuitive behavior of IG is caused by the choice of the baseline. The issue of selecting the proper baseline has been studied in previous work [Kapishnikov, et al., 2019, Srinivas, et al., 2019, Sturmfels, et al., 2020, Pan, et al., 2021,Wang, et al., 2021].
>
> As the choice of the baseline determines the integration path of IG, **the baseline selection can be regarded as determining** **(1) which linear regions are traversed by the path, and (2) how much portion of the path is included in each selected linear region.** However, it is non-trivial to select proper linear regions and adjust the weight of each linear region with only changing the baseline.
>
> Instead of finding the good baseline, we consider the exploration of a sequence of linear regions to generate reliable attribution to interpret the decision of the network. To this end, we propose the distillation process, which select a sequence of linear regions based on the intermediate local attribution.
>
> For the detailed discussion, please refer to our common response.
>
> [Kapishnikov, et al., 2019] Kapishnikov, Andrei, et al. "Xrai: Better attributions through regions." *Proceedings of the IEEE/CVF International Conference on Computer Vision*. 2019.
>
> [Srinivas, et al., 2019] Srinivas, Suraj, and François Fleuret. "Full-gradient representation for neural network visualization." *Advances in neural information processing systems* 32 (2019).
>
> [Sturmfels, et al., 2020] Sturmfels, Pascal, Scott Lundberg, and Su-In Lee. "Visualizing the impact of feature attribution baselines." *Distill* 5.1 (2020): e22.
>
> [Pan, et al., 2021] Pan, Deng, Xin Li, and Dongxiao Zhu. "Explaining Deep Neural Network Models with Adversarial Gradient Integration." *IJCAI*. 2021.
>
> [Wang, et al., 2021] Wang, Zifan, Matt Fredrikson, and Anupam Datta. "Robust Models Are More Interpretable Because Attributions Look Normal." (2021).
>
> ---
>
> ### [Q6] Additional References for Section 3.1
>
> The counter-intuitive behavior of IG is addressed in previous work [Pan, et al., 2021, Kapishnikov, et al., 2019, Srinivas, et al., 2019] with various examples. In particular, FullGrad points out that such behavior can be caused as considering the multiple linear regions, and it is described as violating the weak dependency (local attribution). Although our observation is similar, we try to describe the baseline selection is main cause of the counter-intuitive behavior of IG, and the problem of baseline selection can be rephrased as (1) which linear regions are selected, and (2) how much weight is assigned to each linear region.
>
> To best our knowledge, the vulnerability observed in FullGrad is our original observation. We think that this problem can be caused from handling a single linear region to compute attribution. This would be interpreted as the vulnerability of the local attribution, which measures the attribution using a single point. DeepLIFT also addresses the similar vulnerability of the local attribution when the attribution is computed without considering how the output behaves over a range of inputs [Shrikumar, et al., 2017].
>
> [Kapishnikov, et al., 2019] Kapishnikov, Andrei, et al. "Xrai: Better attributions through regions." *Proceedings of the IEEE/CVF International Conference on Computer Vision*. 2019.
>
> [Srinivas, et al., 2019] Srinivas, Suraj, and François Fleuret. "Full-gradient representation for neural network visualization." *Advances in neural information processing systems* 32 (2019).
>
> [Pan, et al., 2021] Pan, Deng, Xin Li, and Dongxiao Zhu. "Explaining Deep Neural Network Models with Adversarial Gradient Integration." *IJCAI*. 2021.
>
> [Shrikumar, et al., 2017] Shrikumar, Avanti, Peyton Greenside, and Anshul Kundaje. "Learning important features through propagating activation differences." *International conference on machine learning*. PMLR, 2017.

---

> ### Author Response · Authors · 2022-08-02
> **Official response to Reviewer H2WE (2/3)**
>
> ### [Q3] Blurb on the selecting methods to compare
>
> We mainly focus on the comparison among the gradient-based attribution methods, as the DGA is basically a gradient-based method. However, we agree that the quantitative comparison for other types of attribution methods can emphasize the strength of DGA. We attach LeRF and MoRF results for the suggested attribution methods.
>
> The additionally evaluated attribution methods are listed below:
>
> - DeepLIFT
> - LRP
> - SmoothGrad
> - RISE
> - Grad-CAM
>
> From the below table, we can identify that the DGA shows the superior LeRF and MoRF performance for each structure of network. We note that LRP for Inception-V3 is non-trivial to be applied directly due to its specific structure of Inception blocks. We appreciate for the comment that these results would help to strengthen our contribution.
>
> ### LeRF (Higher $\uparrow$ is better)
>
> $\begin{array}{l|ccccccc}
>  & \text{DeepLIFT} & \text{LRP} & \text{SmoothGrad} & \text{RISE} & \text{Grad-CAM} & \text{DGA} \newline\hline
> \text{VGG-16} & 0.095& 0.240 & 0.360 & 0.393 & 0.414 & \textbf{0.434} \newline
> \text{ResNet-18} & 0.143 & 0.348 & 0.375 & 0.410 & 0.429 & \textbf{0.533} \newline
> \text{Inception-V3} & 0.159 & - & 0.548 & 0.500 & 0.554 & \textbf{0.691}
> \end{array}$
>
> ### MoRF (Lower $\downarrow$ is better)
>
> $\begin{array}{l|ccccccc}
>  & \text{DeepLIFT} & \text{LRP} & \text{SmoothGrad} & \text{RISE} & \text{Grad-CAM} & \text{DGA} \newline\hline
> \text{VGG-16} & 0.027 & 0.045 & 0.064 & 0.130 & 0.111 & \textbf{0.023} \newline
> \text{ResNet-18} & 0.041 & 0.062 & 0.078 & 0.115 & 0.115 & \textbf{0.019} \newline
> \text{Inception-V3} & 0.109 & - & 0.114 & 0.144 & 0.123 & \textbf{0.041}
> \end{array}$
>
> [DeepLIFT] Shrikumar, Avanti, Peyton Greenside, and Anshul Kundaje. "Learning important features through propagating activation differences." *International conference on machine learning*. PMLR, 2017.
>
> [LRP] Montavon, Grégoire, et al. "Explaining nonlinear classification decisions with deep taylor decomposition." *Pattern recognition* 65: 211-222. 2017.
>
> [SmoothGrad] Smilkov, Daniel, et al. "Smoothgrad: removing noise by adding noise." *arXiv preprint arXiv:1706.03825.* 2017.
>
> [Grad-CAM] Selvaraju, Ramprasaath R., et al. "Grad-cam: Visual explanations from deep networks via gradient-based localization." *Proceedings of the IEEE international conference on computer vision*. 2017.
>
> [RISE] Petsiuk, Vitali, Abir Das, and Kate Saenko. "Rise: Randomized input sampling for explanation of black-box models." *Proceedings of the British Machine Vision Conference (BMVC)*. 2018.
>
> ---
>
> ### [Q4] Baseline selection for IG
>
> We used the zero-valued input as the baseline for IG, which takes the pixel value of [0.4850, 0.4560, 0.4060] before normalization that widely used for ImageNet. As reviewer pointed out [Sturmfels, et al., 2020], the attribution differs according to the different baseline selection.
> We note that changing the baseline gives the trade-off among several examples. For example, white baseline would give better attribution to dumbbells, but it does not help to attribute the white classes, such as samoyed.
>
> We provide the pixel flip evaluation in the below tables with various baseline selection. We choose following variants as baselines:
>
> - Static baselines
>     - Black image (IG-B)
>     - White image (IG-W)
>     - Black image + White image [Kapishnikov, 2019] (IG-BW)
> - Input dependent baselines
>     - Average pixel value (IG-Avg)
>     - Blurred image
>
> Tables show the pixel flip evaluation results with various baselines. We confirm that the proposed method achieves higher performance compared to IG with various baselines. We appreciate for the comment that these results would help to strengthen our contribution.
>
> ### LeRF (Higher $\uparrow$ is better)
>
> $\begin{array}{l|cccc|cc|c}
> & \text{IG} & \text{IG-B} & \text{IG-W} & \text{IG-BW} & \text{IG-Avg} & \text{IG-Blur} &\text{DGA}\newline\hline
> \text{VGG-16} & 0.096 & 0.070 & 0.073 & 0.080 & 0.079 & 0.071 & \textbf{0.434} \newline
> \text{ResNet-18} & 0.158 & 0.103 & 0.108 & 0.135 & 0.139 & 0.124 & \textbf{0.533} \newline
> \text{Inception-V3} & 0.243 & 0.151 & 0.176 & 0.215 & 0.218 & 0.219 & \textbf{0.691}
> \end{array}$
>
> ### MoRF (Lower $\downarrow$ is better)
>
> $\begin{array}{l|cccc|cc|c}
> & \text{IG} & \text{IG-B} & \text{IG-W} & \text{IG-BW} & \text{IG-Avg} & \text{IG-Blur} &\text{DGA}\newline\hline
> \text{VGG-16} & 0.036 & 0.107 & 0.079 & 0.043 & 0.034 & 0.031 & \textbf{0.023} \newline
> \text{ResNet-18} & 0.038 & 0.078 & 0.057 & 0.035 & 0.040 & 0.035 & \textbf{0.019} \newline
> \text{Inception-V3} & 0.066 & 0.177 & 0.120 & 0.071 & 0.073 & 0.083 & \textbf{0.041}
> \end{array}$
>
> [Sturmfels, et al., 2020] Sturmfels, Pascal, Scott Lundberg, and Su-In Lee. "Visualizing the impact of feature attribution baselines." *Distill* 5.1 (2020): e22.
>
> [Kapishnikov, et al., 2019] Kapishnikov, Andrei, et al. "Xrai: Better attributions through regions." *ICCV*. 2019.

---

> ### Author Response · Authors · 2022-08-02
> **Official response to Reviewer H2WE (1/3)**
>
> We sincerely appreciate your time and efforts in reviewing our paper, as well as the constructive comments and delicate corrections on typos . We respond to each of your questions one by one. All the responses will be carefully incorporated in the final draft.
>
> ---
>
> ### [Q1] Reliability of evaluation without the ground truth
>
> As reviewer addressed, the evaluation of the attribution method is still **challenging due to the absence of the ground truth**. Even if we have the dataset which is labeled with semantics that contributes to the prediction, **it is not certain that the model would learn to predict in such a way that the dataset provides**.
>
> To formulate this issue, attribution functions can be represented as a mapping from the tuple of the model function space $F$ and the input space $\mathbb{R}^d$ to the attribution map $\mathbb{R}^d$, $\phi:F \times \mathbb{R}^d \mapsto \mathbb{R}^d$. In this respective, the ground truth for the attribution should be given as a tuple of the function, the input and the corresponding attribution map $(f, x, \phi(x;f))$. In our best knowledge, such dataset for the deep models are absent.
>
> Rather than the dataset-based evaluation, previous work aims to **evaluate if the attribution method appropriately reflects the behavior of the model prediction**. As noted in the manuscript, Pixel flip evaluates the local behavior on the single instance prediction and ROAR evaluates the identification of features utilized in the training phase. We will address this issue in detail at the beginning of the experiments.
>
> ---
>
> ### [Q2] FullGrad in DGA is replaceable?
>
> The proposed DGA is composed of (1) the local attribution method (e.g., FullGrad), and (2) distillation process which induces the exploration for the multiple linear regions to retain the property of the global attribution method such as IG. So other local attribution methods (e.g., Grad$\times$Input (G\*I)) can also be applied in our distillation process and be improved. We selected FullGrad, because it is known as one of the local attribution methods which utilizes the bias gradient to generate the reliable attribution heatmap.
>
> The below tables show the LeRF and MoRF performance between G\*I with/without the proposed distillation process. From the tables, we empirically verify that the proposed distillation process can improve the performance of G\*I in almost cases. However, DGA which adopts FullGrad as the local attribution method outperforms in both LeRF and MoRF. We note that the red/blue color indicates the increase/decrease of scores from G\*I to DGA(G\*I). We appreciate for the comment that these results would help to strengthen our contribution.
>
> ### LeRF (Higher $\uparrow$ is better)
>
> $\begin{array}{l|cc|c}
> & \text{G\*I} & \text{DGA} (\text{G\*I}) &\text{DGA} \newline\hline
> \text{VGG-16} &  0.078 & 0.420 \scriptsize{\color{red}{\ (+0.342)}} & \textbf{0.434} \newline
> \text{ResNet-18} & 0.171& 0.506 \scriptsize{\color{red}{\ (+0.335)}} & \textbf{0.533} \newline
> \text{Inception-V3} & 0.114 & 0.670 \scriptsize{\color{red}{\ (+0.556)}} & \textbf{0.691}
> \end{array}$
>
> ### MoRF (Lower $\downarrow$ is better)
>
> $\begin{array}{l|cc|c}
> & \text{G\*I} & \text{DGA} (\text{G\*I})  &\text{DGA}\newline\hline
> \text{VGG-16} & 0.045 & 0.028 \scriptsize{\color{blue}{\ (-0.017)}}& \textbf{0.023} \newline
> \text{ResNet-18} &  0.105 & 0.028 \scriptsize{\color{blue}{\ (-0.077)}}  & \textbf{0.019} \newline
> \text{Inception-V3} & 0.050 & 0.066  \scriptsize{\color{red}{\ (+0.016)}} & \textbf{0.041}
> \end{array}$

---

> > ### Comment · Reviewer_H2WE · 2022-08-09
> > **Response to Author Response for Q2**
> >
> > Can you elaborate your reason for using FullGrad? You say it is because it uses the bias gradient. Why, theoretically, did you choose a method that uses the bias gradient? Other than the bias gradient, are there other desirable qualities of an explanation method which would make for effective distillation?
> >
> > On another note, I see that the distillation method generally improves G*I. This is a testimony to the effectiveness of the distillation process. However, the single comparison, while suggestive, seems inconclusive.

---

> > > ### Author Response · Authors · 2022-08-09
> > > **Response to Reviewer H2WE**
> > >
> > > According to the FullGrad original paper [Srinivas, et al., 2019], the exact decomposition of the ReLU network output is represented in terms of the input-gradients and the bias-gradients as below:
> > > $$f(\textbf{x};\textbf{b}) = \nabla\_\textbf{x}f(\textbf{x};\textbf{b})^T \textbf{x} + \nabla\_\textbf{b} f(\textbf{x};\textbf{b})^T \textbf{b}.$$
> > >
> > > We note that considering the input-gradients term soley (i.e., Grad$*$Input) would induce the leakage of information (bias-gradients term) to interpret the output of the network. It indicates that handling bias-gradients can support generating the reliable local attribution.
> > >
> > > In a geometrical view, as both (1) the weight, and (2) the bias determine the trained decision boundary to predict the class, we should consider both information to compute the local attribution for the reliability in interpreting the decision of the network. For these reasons, we adopt FullGrad as the local attribution for our distillation process.
> > >
> > > We are certain that the improvement of G$*$I with the distillation process comes from considering the multiple linear regions to determine final attributions. To verify the effectiveness of the distillation process, we also provide the qualitative comparison for the original FullGrad with distillation process (please see Figure 1 in Section A of supplementary material). $FG+\text{X}$ denotes the original FullGrad with elements of the distillation process. For example, in the third example (airship) with VGG-16 case, we confirm that $FG+WC+EPC$ can generate more object-aligned attributions compared to vanilla FG.
> > >
> > > As a result, we believe that our distillation process which efficiently considers the multiple linear regions can improve the arbitrary local attribution to reflect the behavior of the network in a global sense, which supports the reliable explanation.
> > >
> > > Thank you for the valuable suggestions and comments. We hope this response would address your concerns.
> > >
> > > If you have any remaining suggestions or concerns, please let us know!
> > > Best, Authors.
> > >
> > > [Srinivas, et al., 2019] Srinivas, Suraj, and François Fleuret. "Full-gradient representation for neural network visualization." Advances in neural information processing systems 32 (2019).

---

### Official Review · Reviewer_cvDy · 2022-07-07

**Rating:** 7
**Confidence:** 4
**Soundness:** 4 excellent
**Presentation:** 4 excellent
**Contribution:** 3 good

**Summary:**

This manuscript proposes a new attribution based DNN explanation method that distill input features using weak and extremely positive contributor masks. This Distilled Gradient Aggregation (DGA) approach combines the advantages of both global IG and local FG approaches via a new sequential feature distillation algorithm, distilling irrelevant features from the input. Authors performed quantitative and qualitative evaluation by comparing to various attribution methods.

**Questions:**

Have you discussed the pros and cons of your iterative tuning approach to simple baseline such as saliency map in terms of cost-effectiveness?

**Limitations:**

Just like other methods discussed in this manuscript, it is a post hoc explanation method that attempting to explain a trained model’s output. It does not address the issue of model selection/training for improving explanation quality. Since pre-trained models are used, so the comparison is fair.

Other limitations are time and tuning complexity. It can be cost-prohibitive to explain each single image.

**Strengths And Weaknesses:**

Strengths: The DGA approach effectively addresses the limitations of local approach (FullGrad) that uses a single anchor point to compute attribution. It also addresses counter-intuitive behavior of global approach (IG) with arbitrary integral path. Authors also evaluated performance using ROAR and LeRF/MoRF and demonstrated superior performance. The overall idea of aggregating local attributions via knowledge distillation is novel and experimental evaluations are sound and adequate.

Weaknesses: (1) Authors could discuss the robustness of your explanation approaches against input perturbation since it is a known problem of IG type of methods. (2) Author could perform an evaluation on the percentage of input ablation on model explaining. (3) The sequential distillation process can be time consuming considering the explanation is on per sample basis.

---

> ### Author Response · Authors · 2022-08-02
> **Official response to Reviewer cvDy**
>
> We sincerely appreciate your time and efforts in reviewing our paper, as well as the constructive comments. We respond to your comment below. All the responses will be carefully incorporated in the final draft.
>
> ---
>
> ### [Q1] Evaluation on the percentage of input ablation
>
> To quantify the relation between the ablated input features and computed attribution, we think that sensitivity-$n$ [Ancona, et al., 2018] can be one option. Sensitivity-$n$ has suggested to quantify generalization of the properties of *Completeness* [Sundararajan et al., 2017] and *Summation to Delta* [Shrikumar et al., 2017].  In general, since not all deep learning models can satisfy Sensitivity-$n$, we use experimental methods to determine whether there is an algorithmic bias in the process of calculating attribution. To measure how much the attribution method satisfies this properties, this metric quantifies that the alignment between (1) the sum of the attributions ($\sum_{i\in S}\phi_i(\textbf{x})$) for any subset of features ($S$), and (2) the change of the model output when the input subset is ablated ($\textbf{x}_{S^c}$). The metric can be represented as
>
> $$
>  Corr\bigg[\sum_{i\in S}\phi_i(\textbf{x}), f(\textbf{x}) - f(\textbf{x}_{S^c})\bigg],
> $$
>
> where the subset $S$ is uniform randomly sampled with its cardinality $|S|=n$. **The higher correlation value indicates that the attribution method empirically satisfies the sensitivity-$n$.**
>
> For computational efficiency, instead of ablating individual input features ($n$), we perform patch-wise ablation to compute sensitivity-$n$. To select the patch, we randomly sample the binary mask $M_q\in\mathbb{R}^{14\times14}$ (100 masks for each input) where the portion of 1 in each mask is constrained be $q$. After mask selection, we upsample the mask and multiply the upsampled mask to the attribution and the input to compute the correlation. For the given percentage $q$,  the patch-wise sensitivity can be
>
> $$
>  Corr\bigg[\big\langle upsample(M_q), \phi(\textbf{x})\big\rangle, f\big(\textbf{x}\big) - f\big((1-upsample(M_q)) \odot \textbf{x}\big)\bigg].
> $$
>
> The below table shows the correlation values for various percentage of selection ($q$) over each attribution method for ResNet-18. We identify that DGA has the highest correlation values in entire cases. We appreciate for the comment that these results would help to strengthen our contribution.
>
> $\begin{array}{r|ccccccccc}
> q\\% & 10 & 20 & 30 & 40 & 50 & 60 & 70 & 80 & 90 \newline\hline
> \text{G*I} & -0.006 & -0.009 & -0.009 & -0.013 & -0.007 & -0.014 & -0.016 & -0.022 & -0.037 \newline
> \text{GBP} & 0.022 & 0.032 & 0.025 & 0.024 & 0.022 & 0.027 & 0.024 & 0.024 & 0.020 \newline
> \text{IG} & 0.013 & 0.022 & 0.020 & 0.027 & 0.032 & 0.037 & 0.039 & 0.047 & 0.067 \newline
> \text{GIG} & 0.006 & 0.007 & 0.004 & 0.002 & 0.003 & 0.002 & 0.002 & 0.002 & -0.001 \newline
> \text{FG} & -0.045 & -0.017 & -0.005 & 0.001 & 0.003 & 0.003 & 0.008 & 0.008 & 0.004 \newline
> \text{DGA} & \textbf{0.095} & \textbf{0.096} & \textbf{0.101} & \textbf{0.098} & \textbf{0.095} & \textbf{0.089} & \textbf{0.083} & \textbf{0.080} & \textbf{0.079}
> \end{array}$
>
> [Ancona, et al., 2018] Ancona, Marco, et al. "Towards better understanding of gradient-based attribution methods for Deep Neural Networks." *International Conference on Learning Representations*. 2018.
>
> [Sundararajan, et al., 2017] Sundararajan, Mukund, Ankur Taly, and Qiqi Yan. "Axiomatic attribution for deep networks." *International conference on machine learning*. PMLR, 2017.
>
> [Shrikumar, et al., 2017] Shrikumar, Avanti, Peyton Greenside, and Anshul Kundaje. "Learning important features through propagating activation differences." *International conference on machine learning*. PMLR, 2017.

---

> > ### Comment · Reviewer_cvDy · 2022-08-09
> > **A novel DNN interpretation approach via aggregating local attributions**
> >
> > I would like to thank authors for their time and efforts to address my concerns and others. I expect this work become an important piece in the body of literature on DNN interpretation. As a result, the quality of this manuscript has been increased substantially. I am glad to increase my score to align with the increased quality.

---

> > > ### Author Response · Authors · 2022-08-09
> > > **Thank you for the response**
> > >
> > > We are happy to hear that our rebuttal addressed your concerns well.
> > >
> > > Thank you again for the valuable suggestions and comments to add, which we believe strengthen our paper.
> > >
> > > Best, Authors.

---

### Official Review · Reviewer_hnze · 2022-07-08

**Rating:** 6
**Confidence:** 4
**Soundness:** 2 fair
**Presentation:** 3 good
**Contribution:** 2 fair

**Summary:**

The authors propose a new gradient-based feature attribute method Distilled Gradient Aggregation (DGA) that extends FullGrad (FG). On quantitative and qualitative evaluations the authors demonstrate that their method produces better explanations than previously proposed methods.

**Questions:**

See "Weaknesses" section.

**Limitations:**

The authors briefly discuss the limitations of their work in their conclusion section. They do not address potential negative social impact, which is fine (I don't see any obvious negative impacts with this work).

**Strengths And Weaknesses:**

Overall, I think the ideas in this work are interesting. However, as-is I have significant concerns with the evaluation of the method that prevent me from recommending acceptance. If the authors are able to address my concerns with additional experimental results I would be happy to raise my score.

Strengths:

1. The authors tackle a high-impact problem space (i.e., interpreting neural network predictions).
2. The authors' proposed method is, to my knowledge, novel.
2. Both qualitative and quantitative experimental results indicate that the method produces higher-quality attributions than previously proposed methods.

Weaknesses (in order of importance to my score):

1. Lack of ablations: The authors' proposed method is comprised of multiple moving parts (e.g. the redefined post-processing function $\Phi$, the masking functions, and the aggregation of multiple attribution maps), and it's not clear to me exactly which component(s) led to the improvements over previous methods. For example, based on the data provided in Figure 8 in the supplement, it appears that DGA without the EPC mask (i.e., the purple line with $q=1$), already outperforms many of the baseline models. In my view, the manuscript would be much stronger if additional ablation results were included. On a similar note, it would further strengthen the paper if the authors demonstrated that the individual pieces of their method can't trivially be applied to previous work to obtain similarly strong results. For example, how does the performance of DGA compare to applying integrated gradients with an EPC mask?
2. Influence of the ReLU in aggregation: The authors propose to apply a ReLU function to the individual attribution maps before aggregating them, with the justification of only wanting to retain features that push the network _towards_ a given prediction (rather than away). However, the application of the ReLU function makes the comparison with baseline methods unfair. For example, qualitatively (e.g. in Figure 5), we can see that many of the features with negative attribution values from baseline methods lie in the object of interest of the image (e.g. on the dog's face for the dog image). As such, it's not surprising that DGA performs much better than baselines on the LeRF metric, since the ReLU in DGA would set such attributions to 0. Do these results change if e.g. ReLU is applied to the Integrated Gradients attributions before computing LeRF values?
3. Axiomatic justification (or lack thereof): Many of the previously proposed methods cited in this work (Integrated Gradients, FullGrad, GIG) are guaranteed to satisfy various desirable axioms for attributions. Could the authors comment on which, if any, of the axioms satisfied by previous methods are also satisfied by DGA? If none, is DGA guaranteed to satisfy another set of axioms?
4. Not self-contained: The authors' proposed method is an extension of FullGrad, yet the authors do not provide any background on FullGrad in their manuscript. As such, I frequently needed to refer to the original FullGrad manuscript to be able to understand this work.

---

> ### Author Response · Authors · 2022-08-02
> **Official response to Reviewer hnze (3/3)**
>
> ### [Q3] Axiomatic justification
>
> Among the existing axioms, we provide the analysis on (1) Dummy, (2) Implementation invariance and (3) Sensitivity-$n$.
>
> (1) Dummy
>
> The attribution method $\phi(\cdot)$ satisfies dummy axiom if the function implemented by the deep model does not depend (mathematically) on some variable, then the attribution to that variable is always zero. We note that as the proposed method is an aggregation of the local attribution methods, it satisfies the dummy axiom if the local attribution satisfies. As FullGrad is built upon the gradients, this property can be shown using the gradient. If the deep model does not depend on some variable $\text{x}_i$, then $\frac{df(\textbf{x})}{d\text{x}_i}=0$, which gives the zero attribution. Thus, our method satisfies the dummy axiom.
>
> (2) Implementation invariance
>
> Let two deep models $f_1$ and $f_2$ be functionally equivalent, such that $\forall x,\ f_1(x)=f_2(x)$. Assume that their implementations are different, so that $\exists x,\ h_1(x)\neq h_2(x)$, where $h_i(.)$ represents the hidden layer output of $f_i(.)$. The attribution method satisfies implementation invariance if the computed attribution is same for the functionally equivalent models, $f_1$ and $f_2$.
>
> As current version of DGA adapts a variant of FullGrad,  it does not satisfy the implementation invariance because it utilizes the bias gradient $\frac{df_i}{dh_i}$, where $\frac{df_1}{dh_1}\neq\frac{df_2}{dh_2}$. However, if we use the Grad$\times$Input as the local attribution with distillation process, it satisfies the implementation invariance because $\frac{df_i(x)}{dx}=\frac{df_i(x)}{dh_i(x)}\frac{dh_i(x)}{dx}$ stays same by the chain rule (for the comparison of DGA with Grad$\times$Input, please refer to the response of Q2 to Reviewer H2WE).
>
> (3) Sensitivity-$n$ [Ancona, et al., 2018]
>
> Sensitivity-$n$ has suggested to quantify generalization of the properties of *Completeness* [Sundararajan et al., 2017] and *Summation to Delta* [Shrikumar et al., 2017]. In general, since not all deep learning models can satisfy Sensitivity-$n$, we use experimental methods to determine whether there is an algorithmic bias in the process of calculating attribution. To measure how much the attribution method satisfies this properties, we can empirically quantify the alignment between (1) the sum of the attributions ($\sum_{i\in S}\phi_i(\textbf{x})$) for any subset of features ($S$), and (2) the change of the model output when the input subset is ablated ($\textbf{x}_{S^c}$). The metric can be represented as
>
> $$
>  Corr\bigg[\sum_{i\in S}\phi_i(\textbf{x}), f(\textbf{x}) - f(\textbf{x}_{S^c})\bigg],
> $$
>
> where the subset $S$ is uniform randomly sampled with its cardinality $|S|=n$. The higher correlation value indicates that the attribution method empirically satisfies the sensitivity-$n$.
>
> Below table shows the empirical evaluation of sensitivity-$n$ to compare various attribution methods. For computational efficiency, we test on the $q\\%$ of image randomly ablated to evaluate. We confirm that our method shows the highest correlation. As a result, we empirically verify that DGA is likely to satisfy sensitivity-$n$ among various attribution methods.
>
> $\begin{array}{r|ccccccccc}
> q\\% & 10 & 20 & 30 & 40 & 50 & 60 & 70 & 80 & 90 \newline\hline
> \text{G*I} & -0.006 & -0.009 & -0.009 & -0.013 & -0.007 & -0.014 & -0.016 & -0.022 & -0.037 \newline
> \text{GBP} & 0.022 & 0.032 & 0.025 & 0.024 & 0.022 & 0.027 & 0.024 & 0.024 & 0.020 \newline
> \text{IG} & 0.013 & 0.022 & 0.020 & 0.027 & 0.032 & 0.037 & 0.039 & 0.047 & 0.067 \newline
> \text{GIG} & 0.006 & 0.007 & 0.004 & 0.002 & 0.003 & 0.002 & 0.002 & 0.002 & -0.001 \newline
> \text{FG} & -0.045 & -0.017 & -0.005 & 0.001 & 0.003 & 0.003 & 0.008 & 0.008 & 0.004 \newline
> \text{DGA} & \textbf{0.095} & \textbf{0.096} & \textbf{0.101} & \textbf{0.098} & \textbf{0.095} & \textbf{0.089} & \textbf{0.083} & \textbf{0.080} & \textbf{0.079}
> \end{array}$
>
> [Ancona, et al., 2018] Ancona, Marco, et al. "Towards better understanding of gradient-based attribution methods for Deep Neural Networks." *International Conference on Learning Representations*. 2018.
>
> [Sundararajan, et al., 2017] Sundararajan, Mukund, Ankur Taly, and Qiqi Yan. "Axiomatic attribution for deep networks." *International conference on machine learning*. PMLR, 2017.
>
> [Shrikumar, et al., 2017] Shrikumar, Avanti, Peyton Greenside, and Anshul Kundaje. "Learning important features through propagating activation differences." *International conference on machine learning*. PMLR, 2017.

---

> > ### Comment · Reviewer_hnze · 2022-08-05
> > **Thank you for the additional results**
> >
> > Dear authors,
> >
> > Thank you for providing the additional experimental results; I believe that these significantly strengthen the manuscript. It was particularly nice to see the results of the new ablation study and to confirm that DGA outperforms IG + ReLU in the LeRF score. As such, I have raised my score. I would strongly encourage the authors to include these new results and to take into account the writing changes suggested by the reviewers for the final version of the manuscript.

---

> > > ### Author Response · Authors · 2022-08-07
> > > **Thank you for the response**
> > >
> > > We are happy to hear that our rebuttal addressed your concerns well.
> > >
> > > Thank you again for the valuable suggestions and comments, which we will incorporate in the final version to further strengthen our paper.
> > >
> > > If you have any remaining suggestions or concerns, please let us know!
> > >
> > > Best, Authors.

---

> ### Author Response · Authors · 2022-08-02
> **Official response to Reviewer hnze (2/3)**
>
> ### [Q2] Influence of ReLU
>
> We additionally evaluate LeRF and MoRF on IG and GIG with ReLU for the fair comparison with the proposed method. We provide two variants of applying ReLU in IG and GIG,
>
> $$\begin{cases}
> ReLU\big(\int_{\alpha=0}^1 \frac{dF(\gamma(\alpha))}{d\gamma_i(\alpha)}\frac{d\gamma_i(\alpha)}{d\alpha}d\alpha\big),\quad&(1)\newline
> \int_{\alpha=0}^1 ReLU\big(\frac{dF(\gamma(\alpha))}{d\gamma_i(\alpha)}\frac{d\gamma_i(\alpha)}{d\alpha}\big)d\alpha,\quad&(2)
> \end{cases}$$
>
> whether **(1) ReLU is applied at the end or (2) during the integration**. $F$ is the model function and $\gamma
> =(\gamma_1,\dots,\gamma_n):[0,1]\mapsto \mathbb{R}^n$ is a path generating function that maps from $[0,1]$ to the input space $\mathbb{R}^n$ [Sundararajan, et al., 2017, Kapishnikov, et al., 2021].
>
> The following table shows the pixel flip evaluation results compared to the vanilla IG, DGA, and variants of IG and GIG. We mark the increase/decrease of the score in red/blue color. As the reviewer pointed out, ReLU function can help to improve the LeRF score for IG and GIG. However, applying ReLU function would degrade the MoRF score. We also note that the proposed method still significantly outperforms in the both LeRF and MoRF evaluation. We appreciate for the comment that these results would help to strengthen our contribution.
>
> ### LeRF (Higher $\uparrow$ is better)
>
> $\begin{array}{l|ccc|ccc|c}
> & \text{IG} & \text{IG+ReLU (1)} & \text{IG+ReLU (2)} & \text{GIG} & \text{GIG+ReLU (1)} & \text{GIG+ReLU (2)} & \text{DGA}\newline\hline
> \text{VGG-16}&0.096&0.141\scriptsize{\color{red}{\ (+0.045)}}&0.262\scriptsize{\color{red}{\ (+0.166)}}&0.110&0.157\scriptsize{\color{red}{\ (+0.047)}}&0.325\scriptsize{\color{red}{\ (+0.215)}}&\textbf{0.434}\newline
> \text{ResNet-18}&0.158&0.204\scriptsize{\color{red}{\ (+0.046)}}&0.339\scriptsize{\color{red}{\ (+0.181)}}&0.185&0.239\scriptsize{\color{red}{\ (+0.054)}}&0.401\scriptsize{\color{red}{\ (+0.216)}}&\textbf{0.533}\newline
> \text{Inception-V3}&0.243&0.233\scriptsize{\color{blue}{\ (-0.010)}}&0.587\scriptsize{\color{red}{\ (+0.344)}}&0.255&0.239\scriptsize{\color{blue}{\ (-0.016)}}&0.610\scriptsize{\color{red}{\ (+0.356)}}&\textbf{0.691}\newline
> \end{array}$
>
> ### MoRF (Lower $\downarrow$ is better)
>
> $\begin{array}{l|ccc|ccc|c}
> & \text{IG} & \text{IG+ReLU (1)} & \text{IG+ReLU (2)} & \text{GIG} & \text{GIG+ReLU (1)} & \text{GIG+ReLU (2)} & \text{DGA}\newline\hline
> \text{VGG-16}&0.036&0.036\scriptsize{\color{gray}{\ (+0.000)}}&0.035\scriptsize{\color{blue}{\ (-0.001)}}&0.029&0.030\scriptsize{\color{red}{\ (+0.001)}}&0.036\scriptsize{\color{red}{\ (+0.007)}}&\textbf{0.023}\newline
> \text{ResNet-18}&0.038&0.041\scriptsize{\color{red}{\ (+0.003)}}&0.039\scriptsize{\color{red}{\ (+0.001)}}&0.029&0.032\scriptsize{\color{red}{\ (+0.003)}}&0.036\scriptsize{\color{red}{\ (+0.007)}}&\textbf{0.019}\newline
> \text{Inception-V3}&0.066&0.125\scriptsize{\color{red}{\ (+0.059)}}&0.070\scriptsize{\color{red}{\ (+0.004)}}&0.061&0.120\scriptsize{\color{red}{\ (+0.059)}}&0.069\scriptsize{\color{red}{\ (+0.008)}}&\textbf{0.041}\newline
> \end{array}$
>
> [Sundararajan, et al., 2017] Sundararajan, Mukund, Ankur Taly, and Qiqi Yan. "Axiomatic attribution for deep networks." *International conference on machine learning*. PMLR, 2017.
>
> [Kapishnikov, et al., 2021] Kapishnikov, Andrei, et al. "Guided integrated gradients: An adaptive path method for removing noise." *Proceedings of the IEEE/CVF conference on computer vision and pattern recognition*. 2021.

---

> ### Author Response · Authors · 2022-08-02
> **Official response to Reviewer hnze (1/3)**
>
> We sincerely appreciate your time and efforts in reviewing our paper, as well as the constructive comments. We respond to each of your questions one by one. We also include the backgrounds on FullGrad so that readers would better understand our work. All the responses will be carefully incorporated in the final draft.
>
> ---
>
> ### [Q1] Ablation study of the proposed method
>
> At first, we briefly introduce the intuitions behind each element of the proposed method, and we share the results of ablation studies for each element in LeRF and MoRF.
>
> (1) Modifying the post-processing in FullGrad
> In FullGrad, the post-processing utilizes the upsampled bias gradient to generate the final attribution. It is known that **the upsampling-based attribution usually cause the low resolution** such as GradCAM. In previous work [Grabska-Barwinska, et al., 2021], we can identify that **the bias gradient is mainly over-estimated in the deeper layers.** To alleviate these two problems, we redesign the post-processing to distribute gradient uniformly. We expect the obtained attribution can have more sharp heatmap compared to the upsampling-based approach.
>
> (2) WC mask
>
> As mentioned the common response, for the global attribution, we should consider multiple linear regions. One option is **removing the weak relevant input features (i.e., pixels) to move the given input to the other linear region.** Weak Contributor (WC) mask is sequentially removing the weak relevant input features depending on the intermediate local attribution. As a result, the given input sequentially moves through linear regions where the perturbed input resides, and finally arrives to the zero-baseline. We expect the local attribution with WC mask can have more reliable attribution to interpret the decision of the network.
>
> (3) EPC mask
>
> As described in the main paper, we empirically observe that the local attributions from the selected linear regions with WC mask are sometimes saturated. It means that some pixels consistently have high relevance, which make the final attributions be over-concentrated. EPC mask is designed to alleviate this biased attribution. We expect that EPC mask reduces the over-concentrated attribution and it helps to improve the performance of LeRF and MoRF, which indicates the reliable explanation for the decision of the network.
>
> The below table shows the quantitative ablation studies for each element using LeRF and MoRF.  The quantitative result indicates that using WC helps to achieve better LeRF score, and using EPC improves MoRF score. To complement the strengths of both approaches, we combine two methods to obtain the marginal scores in LeRF and MoRF. The qualitative comparison is provided in Figure 1 of the supplementary material. We additionally provide the FullGrad with our distillation process to verify the effect of (1) the modifying of post-processing. From the Figure 1, we identify that changed post-processing contributes to improve the resolution of the attribution heatmap.
>
>
>
> ### LeRF (Higher $\uparrow$ is better)
>
> \begin{array}{l|ccccccc}
>  & \text{DGA (WC)} & \text{DGA (EPC)} & \text{DGA (WC+EPC)} \newline\hline
> \text{VGG-16} & 0.417 & 0.349 & \textbf{0.434} \newline
> \text{ResNet-18} & \textbf{0.551} & 0.479 & 0.533 \newline
> \text{Inception-V3} & \textbf{0.719} & 0.646 & 0.691
> \end{array}
>
> ### MoRF (Lower $\downarrow$ is better)
>
> \begin{array}{l|ccccccc}
>  & \text{DGA (WC)} & \text{DGA (EPC)} & \text{DGA (WC+EPC)} \newline\hline
> \text{VGG-16} & 0.039 & \textbf{0.018} & 0.023 \newline
> \text{ResNet-18} &0.034 & \textbf{0.014} & 0.019 \newline
> \text{Inception-V3} & 0.082 & \textbf{0.029} & 0.041
> \end{array}
>
> [Grabska-Barwinska, et al., 2021] Grabska-Barwinska, Agnieszka, et al. "Towards Better Visual Explanations for Deep Image Classifiers." *eXplainable AI approaches for debugging and diagnosis*. 2021.

---

### Author Response · Authors · 2022-08-02
**Common response**

We sincerely appreciate the reviewers’ time and efforts in reviewing our paper, as well as the constructive comments. In this thread, we respond to the commonly asked question: counter-intuitive behavior of IG.

---

We first note that “counter-intuitive behavior of IG” is raised to describe the observation that the attribution computed by IG for input feature $x_2$ is changed when only the input feature $x_1$ is modified, which is represented in Figure 3 of the main paper.

As the reviewers pointed out, the referred problem is caused by the problem of the baseline selection. From the definition of IG for a function $F$ and an input $X$, $\int_{\alpha=0}^1\frac{df(\gamma(\alpha))}{d\gamma(\alpha)}\frac{\gamma(\alpha)}{d\alpha}d\alpha$, the path function $\gamma(\alpha)=\bar{X}-\alpha(X-\bar{X})$ is determined by the baseline $\bar{X}$. That is, **the selection of baseline determines (1) which linear regions are traversed by the path $\gamma$, and (2) how much portion of the path $\gamma$ is included in each selected linear region.**

With the large number of linear regions and the complicated configuration of the linear regions as a result of training deep models, it is hard to guarantee that traversed linear regions by the determined path are always useful to quantify the reliable relevance for the input features. We would like to point out that such undesirable linear regions traversed by the path can induce the counter-intuitive attributions, as shown in the example of Figure 3.

Although the proper selection of baseline can be an one option to resolve (1) selection of traversal linear regions, and (2) weighting for each linear region, it is still non-trivial to control the sequence of the meaningful linear regions and each weight by only changing the baseline.

On the other hand, **the local attribution method (e.g., FullGrad) which considers a single linear region** can be alternative to alleviate the problem of the baseline selection. However, it is hard to interpret the decision of the network in a global perspective (in Figure 2 of the main paper, we observe that Fullgrad can be vulnerable for small gaussian noise), but **global attribution method (e.g., IG) which considers multiple linear regions** is relatively robust.

As a result, instead of finding the good baselines, we consider the exploration of sequence of linear regions to generate reliable attribution to interpret the decision of the network. To this end, we first adopt the local attribution method to alleviate the selection of the undesirable linear regions caused by the baseline selection. Second, to consider multiple linear regions, which can support an explanation with a global perspective for the network, we design a distillation process to select sequence of multiple linear regions based on the intermediate local attribution. Finally, we aggregate these local attributions to determine final attributions.

---

### Author Response · Authors · 2022-08-07
**A gentle reminder**

Thank you very much again for your time and efforts in reviewing our paper.

We kindly remind that we have only few days for the author-reviewer discussion period, which ends this Tuesday.

We wonder whether there is any further concerns and hope to have a chance to respond before the discussion phase ends.

Regards,
Authors

---

### Meta-Review · Area_Chair_reMb · 2022-08-27

**Recommendation:** Accept
**Confidence:** Certain

**Metareview:**

This paper proposes a new gradient-based attribution method Distilled Gradient Aggregation (DGA), which combines the strengths of both local and global attribution methods. The reviewers and meta-reviewer found the method novel, and supported by promising results both qualitatively and quantitatively.

During the rebuttal phase, the authors made a _thorough_ effort in response to each reviewer's comments. As recognized by two reviewers (hnze, cvDy), the newly added results and discussions have significantly strengthened the contribution.

The AC recommends acceptance given the paper tackles a critical problem, and presented an effective and convincing method that advances the field of explainable AI.

Authors are strongly recommended to include these new results and writing changes suggested by the reviewers for the final version of the manuscript.









**Award:**

No

---

### Decision · Program_Chairs · 2022-09-14

Accept